# The solid effect of dynamic nuclear polarization in liquids II: Accounting for g-tensor anisotropy at high magnetic fields

Deniz Sezer, Danhua Dai, and Thomas F. Prisner

Institute of Physical and Theoretical Chemistry, Goethe University, 60438 Frankfurt am Main, Germany

**Correspondence:** Deniz Sezer (dzsezer@gmail.com)

**Abstract.** In spite of its name, the solid effect of dynamic nuclear polarization (DNP) is operative also in viscous liquids, where the dipolar interaction between the polarized nuclear spins and the polarizing electrons is not completely averaged out by molecular diffusion on the time scale of the electronic spin-spin relaxation time. Under such slow-motional conditions, it is likely that the tumbling of the polarizing agent is similarly too slow to efficiently average the anisotropies of its magnetic tensors on the electronic $T_2$ time scale. Here we extend our previous analysis of the solid effect in liquids to account for the effect of g-tensor anisotropy at high magnetic fields. Building directly on the mathematical treatment of slow-tumbling in electron spin resonance (Freed et al., 1971), we calculate solid-effect DNP enhancements in the presence of both translational diffusion of the liquid molecules and rotational diffusion of the polarizing agent. To illustrate the formalism, we analyze high-field (9.4 T) DNP enhancement profiles from nitroxide-labeled lipids in fluid lipid bilayers. By properly accounting for power-broadening and motional-broadening, we successfully decompose the measured DNP enhancements into their separate contributions from the solid and Overhauser effects.

## 1  Introduction

The sensitivity of NMR experiments is greatly increased by dynamic nuclear polarization (DNP),[1] where the much larger static polarization that is available to electronic spins is transferred to nuclear spins (Atsarkin, 2011; Wenckebach, 2016). For the transfer to take place, the electronic and nuclear spins should be able to flip simultaneously (Abragam and Goldman, 1978). Such concerted flips correspond to the zero-quantum (ZQ) and double-quantum (DQ) transitions of the electron-nucleus spin system, which are enabled by the inter-spin interactions. Among the four DNP mechanisms, namely the Overhauser effect (OE), the solid effect (SE), the cross effect, and thermal mixing, only the first two have been conclusively shown to be operative also in the liquid state where the spin-spin interactions change randomly in time due to the thermal motions of the molecules.

In OE-DNP, the ZQ and DQ transitions are in fact possible because the dipole-dipole and contact interactions are modulated by the molecular motions. In SE-DNP, on the other hand, the ZQ and DQ transitions are driven directly by the microwave (mw)

---

[1] Abbreviations used in the text: continuous wave (cw), 1,2-dioleoyl-sn-glycero-3-phosphocholine (DOPC), double quantum (DQ), dynamic nuclear polarization (DNP), electron paramagnetic resonance (EPR), force-free hard sphere (FFHS), microwave (mw), nuclear magnetic resonance (NMR), Overhauser effect (OE), 1-palmitoyl-2-stearoyl-sn-glycero-3-phosphocholine (PSPC), solid effect (SE), stochastic Liouville equation (SLE), zero quantum (ZQ).

excitation, and the modulation of the dipolar interaction is detrimental because it constantly modifies the matching condition that the mw frequency should satisfy in order to resonantly drive these transitions.

The initial theoretical treatments of OE (Solomon, 1955) and SE (Abragam and Proctor, 1958) modeled the ZQ and DQ transitions by expressing the transition probabilities per unit time using Fermi's golden rule. As the mathematical description of (semi-classical) relaxation theory matured around the same time (Redfield, 1957; Abragam, 1961), the Fermi golden rule was promptly replaced in the theory of OE-DNP in liquids (Hausser et al., 1968) by the correlation function of the dipolar interaction (or its Laplace transform, which is known as spectral density). Because the time-domain description of relaxation leads to a correlation function in a very general way (Abragam, 1961), the same formalism works naturally with different spectral densities (e.g., for rotational or translational diffusion). As an example, the improved analytical treatment of isotropic translational diffusion achieved in 1975 was immediately applied to paramagnetic relaxation in liquids (Ayant et al., 1975; Hwang and Freed, 1975).[2]

During the same time period, it also became possible to account for spin dephasing and relaxation beyond second order (Anderson, 1954; Kubo, 1954), which is important for understanding spectral line shapes outside the regime of fast averaging (Kubo, 1969). These initial ideas were transformed into a powerful tool for the calculation and analysis of slow-motional EPR spectra by Freed (Freed et al., 1971; Freed, 1976).

When first presented, Abragam's quantitative description of SE-DNP in terms of mixing of the Zeeman energy levels by the dipolar interaction (Abragam and Proctor, 1958) conclusively explained that the NMR signal is maximally enhanced when the mw frequency is shifted from the electronic resonance by $\pm\omega_I$, where $\omega_I$ is the Larmor frequency of the polarized nuclear spin. Abragam's perturbative analysis also correctly predicted that the effect should drop quadratically with the magnitude of the static magnetic field, which has lasting implications for SE-DNP at high magnetic fields. In spite of these successes, however, the perturbative approach to SE is practically impossible to integrate with other relevant spin phenomena whose mathematical treatment matured subsequently.

Recently, Sezer (2023a) presented a time-domain description of SE which, like semi-classical relaxation theory, allows for different dynamical processes to modulate the relevant spin interactions. By interfacing this description with the spectral density of isotropic translational diffusion (Ayant et al., 1975; Hwang and Freed, 1975), it was possible to treat SE-DNP in the presence of molecular translation, as relevant to homogeneous liquids (Sezer, 2023b). The requirement that the dipolar interaction should not be completely averaged out by the molecular dynamics during the electronic $T_2$ restricts liquid-state SE-DNP to viscous media, where the tumbling of the polarizing agent may similarly be too slow to average the anisotropies of its magnetic tensors. The current paper accounts for the effect of g-tensor anisotropy on SE in this slow-tumbling regime. To this end, the time-domain description of SE-DNP in liquids is interfaced here with the established mathematical treatment of slow-motional EPR spectra (Freed et al., 1971). For the illustrative purposes of the current paper, we consider only free (i.e., unrestricted) rotational diffusion with an isotropic diffusion coefficient. Nevertheless, the treatment can be analogously

---

[2]Surprisingly, this improved treatment is not mentioned by Müller-Warmuth et al. (1983), who continue to use the older, deficient expression of the spectral density for translational diffusion.

extended to anisotropic diffusion in an orienting potential by building on the general mathematical formalism of the MOMD and SRLS models (Meirovitch et al., 1984; Polimeno and Freed, 1995).

To motivate the presented theoretical analysis, in Sect. 2 we formulate one specific practical problem that it addresses. There we also introduce the experimental EPR and DNP data that are analyzed subsequently in Sect. 5 using the developed theory. The needed background from Sezer (2023a, b) is presented in Sect. 3. Building on it, in Sect. 4 we adapt the slow-motional formalism of Freed et al. (1971) to the treatment of SE in the liquid state. Our conclusions are in Sect. 6, and several supporting figures are left to the appendix.

## 2 Motivation

DNP aims to increase the longitudinal nuclear magnetization, $i_z$, beyond its equilibrium Boltzmann value, $i_z^{\mathrm{eq}}$. This is done by doping the sample with unpaired electrons, whose spins are then subjected to near-resonance microwave irradiation. In continuous-wave (cw) DNP, which is the only variety that we consider here, a steady state magnetization $i_z^{\mathrm{ss}}$ is reached after the microwaves have been applied for sufficiently long time. The enhancement of $i_z$ under such steady-state conditions is

$$\epsilon = \frac{i_z^{\mathrm{ss}}}{i_z^{\mathrm{eq}}} - 1, \tag{1}$$

where $\epsilon = 0$ corresponds to the absence of DNP.

In both OE and SE, $\epsilon$ is directly proportional to the ratio of the gyromagnetic factors of the electronic and nuclear spins, $\gamma_S$ and $\gamma_I$. For OE (Hausser et al., 1968; Müller-Warmuth et al., 1983),

$$\epsilon_{\mathrm{OE}} = scf \frac{|\gamma_S|}{\gamma_I}, \tag{2}$$

where $s$, $c$ and $f$ are, respectively, the electronic saturation factor, the coupling factor and the leakage factor. The former is defined as

$$s = 1 - s_z^{\mathrm{ss}}/s_z^{\mathrm{eq}} \tag{3}$$

and reflects the deviation of the longitudinal electronic magnetization at steady state, $s_z^{\mathrm{ss}}$, from its equilibrium value, $s_z^{\mathrm{eq}}$. The other two factors, $c$ and $f$, quantify the interaction between the electronic and nuclear spins. Specifically, the leakage factor

$$f = 1 - T_{1I}/T_{1I}^0 \tag{4}$$

compares the nuclear $T_1$'s in the presence ($T_{1I}$) and in the absence ($T_{1I}^0$) of the polarizing agent. In DNP, $T_{1I}$ is typically (much) shorter than $T_{1I}^0$ due to the elevated concentration of the electronic spins, hence $f \approx 1$.

Similarly, the SE enhancement can be expressed as (Sezer, 2023a)

$$\epsilon_{\mathrm{SE}} = pv_- T_{1I} \left( \frac{1}{1 + v_+ T_{1I}} \right) \frac{|\gamma_S|}{\gamma_I}, \tag{5}$$

where $p = 1 - s$ quantifies how "non-saturated" the electronic transition is, and the rate constants $v_{\pm}$ are related to the ability of the microwaves to excite simultaneous flips of the electronic and nuclear spins. These concerted flips correspond to the "forbidden" ZQ and DQ transitions, which are enabled by the dipolar interaction. In fact,

$$v_{\pm} = v_2 \pm v_0, \tag{6}$$

where $v_0$ and $v_2$ denote, respectively, the ZQ and DQ transition rates. In liquids, where the dipolar interaction is partially averaged, the contribution of the mw excitation to the nuclear relaxation rate $R_{1I} = 1/T_{1I}$, which is quantified by $v_+$, is generally negligibly small. As a result, $v_+/R_{1I} \ll 1$ and the expression in parenthesis in Eq. (5) is essentially one. Then the SE enhancement acquires the multiplicative form

$$\epsilon_{\mathrm{SE}} \approx p v_- T_{1I} \frac{|\gamma_S|}{\gamma_I} \qquad (v_+ \ll R_{1I}), \tag{7}$$

which is analogous to $\epsilon_{\mathrm{OE}}$ with the factors $s$, $c$ and $f$ being replaced by the factors $p$, $v_-$ and $T_{1I}$, respectively. In the numerical work presented in Sect. 5 we use the approximation in Eq. (7). The condition $v_+ T_{1I} \ll 1$ is validated at the end of the analysis by comparing the estimated $v_+$ to the measured $T_{1I}$.

In the current paper we study the dependence of the DNP enhancement on the displacement from the electronic resonance. Following Gizatullin et al. (2022), we call the profile of $\epsilon$ against the offset from resonance a "DNP spectrum". Because DNP experiments in the liquid state are carried out with a mw resonator (Erb et al., 1958a, b; Leblond et al., 1971b; Neudert et al., 2016; Gizatullin et al., 2021a; Kuzhelev et al., 2022, 2023), off-resonance conditions are achieved by varying the stationary magnetic field at constant mw frequency (i.e., field sweep). In theoretical analysis, however, it is more convenient to work with a fixed $B_0$ and a variable mw frequency. Thus, when comparing calculations and experiments, we will convert the horizontal axis of the experiments from magnetic field to offset frequency.

In the case of $\epsilon_{\mathrm{OE}}$ (Eq. (2)), the entire offset dependence is due to the saturation factor $s$, as the factors $c$ and $f$ are practically constant over such narrow frequency range. In the case of $\epsilon_{\mathrm{SE}}$ (Eq. (5)), both $p v_-$ and $v_+$ are functions of the offset. For a single, homogeneously-broadened EPR line the saturation factor can be obtained in closed analytical form from the Bloch equations (as we review below in Sect. 3.1). Recently Sezer (2023a) showed that the SE spin dynamics is described by two coupled Bloch equations, whose steady state can similarly be solved analytically to obtain closed-form expressions for the rate constants $v_{\pm}$ (reviewed in Sect. 3.2). In liquids, where the random molecular motion modulates the dipolar interaction between the electronic and nuclear spins, these rate constants are no longer available analytically but can be calculated numerically for motional models with known dipolar spectral densities (Sezer, 2023b), as reviewed below in Sect. 3.3.

Liquid-state SE-DNP is restricted to viscous media, where the dipolar interaction is not averaged out completely by the molecular motions on the decoherence time scale of the electronic spins. Under these conditions, the tumbling of the polarizing agent is also expected to be too slow to average the anisotropies of its magnetic tensors on the electronic $T_2$ time scale. One thus expects substantial deviations from the Lorentzian EPR line shape of the Bloch equations. Such deviations are unavoidable in the case of nitroxide-based polarizing agents whose g and A tensors are rather anisotropic. A recent SE-DNP study at 9.4 T demonstrated that even the narrow-line radical trityl exhibited g-tensor broadening in liquid glycerol (Kuzhelev et al., 2023).

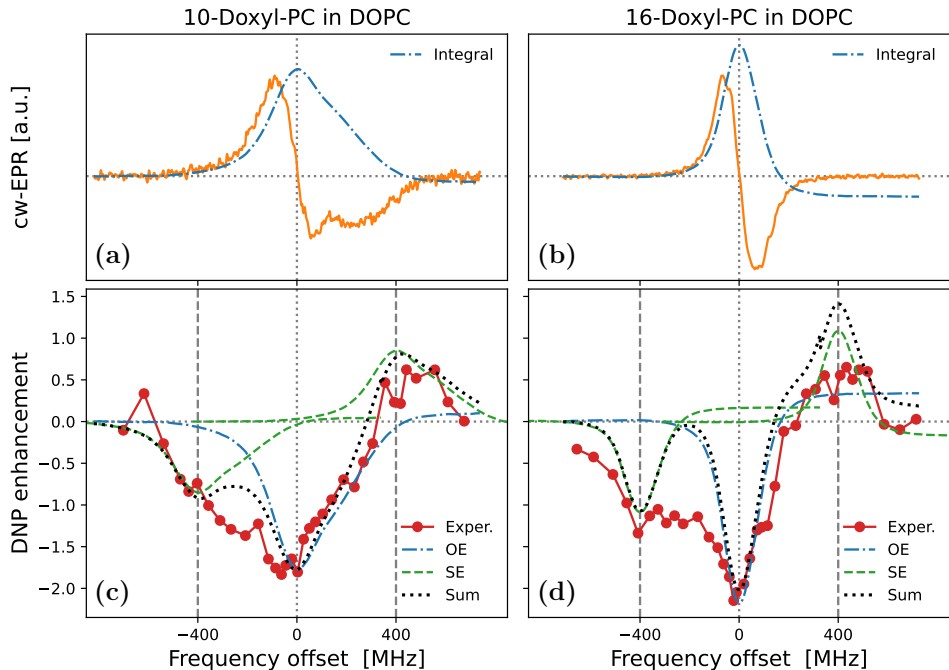

**Figure 1.** Experimental cw-EPR spectra (**a**, **b**) and DNP spectra (**c**, **d**) of spin-labeled lipids in DOPC lipid bilayers at 9.4 T and ≈ 320 K. The nitroxide spin label (Doxyl) is either at position 10 (**a**, **c**) or at position 16 (**b**, **d**) of the aliphatic lipid chain. The integrated cw-EPR spectra (dashed-dotted blue lines in **a** and **b**) are used to decompose the DNP spectra (**c** and **d**) into contributions from OE (dashed-dotted blue lines) and SE (dashed green lines).

This paper extends the theoretical description of SE-DNP to the regime of slow radical tumbling where the cw-EPR line shape in not Lorentzian. Given our longstanding efforts in liquid-state DNP at 9.4 T, here we focus on high magnetic fields, where the width of the EPR spectrum is dominated by the anisotropy of the g tensor. We will thus completely neglect the hyperfine tensor. This possibility greatly simplifies the needed adjustments to the Lorentzian case (Sect. 4).

To illustrate the practical problem that motivated this theoretical work, we now turn to the experimental data in Fig. 1. The characterized samples comprised liposomes of hydrated lipid bilayers composed of DOPC (1,2-dioleoyl-sn-glycero-3-phosphocholine) lipids. As the phase transition temperature of DOPC is about $-17\,°C$, the lipids were in their fluid, liquid-crystalline phase in the experiments at $≈ 320\,K$. The DOPC lipids were mixed at a ratio of 20:1 with PSPC lipids spin-labeled either at position 10 (1-palmitoyl-2-stearoyl-(10-doxyl)-sn-glycero-3-phosphocholine) or at position 16 along one of their aliphatic chains. Both the EPR spectra (Fig. 1a,b) and the DNP enhancements (Fig. 1c,d) were recorded in our home-built Fabry-Pérot resonator at 9.4 T, equipped with temperature control (Denysenkov et al., 2022). While the target temperature of the experiments was 320 K, an extra temperature rise of less than $10\,°C$ can be expected at the maximum mw power of 5.5 W that was used for DNP (Denysenkov et al., 2022). Details about the experiments and the sample preparation will be published elsewhere.

The cw-EPR spectrum of 10-Doxyl-PC in Fig. 1a (orange line) is seen to deviate substantially from (the derivative of) a Lorentzian line shape. At this high magnetic field, the EPR line width is expected to be dominated by the large anisotropy of the nitroxide g tensor, with comparatively much smaller contribution from the nitroxide hyperfine tensor. (These expectations are tested and verified below in Sect. 5.1.) For comparison, in Fig. 1b we show the cw-EPR spectrum of the sample doped with 16-Doxyl-PC. Visually, this narrower spectrum more closely resembles a homogeneous Lorentzian line, although it still deviates from it (as discussed in Sect. 5.2).

In Fig. 1c we show the DNP spectrum (filled red circles) of the sample containing 10-Doxyl-PC as a polarizing agent. The enhanced NMR signal belongs to the acyl chain protons of the lipids. Thanks to the high magnetic field of the experiment, it was possible to resolve the NMR signal of these non-polar protons from the polar protons of water and of the lipid head groups. The DNP spectrum is seen to have a complex line shape, with the positive enhancement values at offsets of about $+400\,\mathrm{MHz}$ demonstrating contribution from SE. At the same time, the comparatively larger negative enhancements in the vicinity of the electronic resonance (i.e., around $0\,\mathrm{MHz}$) point to a contribution from OE. Such coexistence of SE and OE is well documented for nitroxide free radicals at the classical EPR fields of about 0.35 T (Leblond et al., 1971b; Neudert et al., 2017; Gizatullin et al., 2021a, b). Evidently, it also persists at 9.4 T. The DNP spectrum of 16-Doxyl-PC in Fig. 1d also exhibits a mixture of SE and OE.

More than half a century ago Korringa and coworkers developed a rigorous theoretical framework to predict such mixed DNP spectra in viscous liquids (Papon et al., 1968; Leblond et al., 1971a). Likely because of its complexity, as well as its neglect of translational diffusion, their formal analysis has not been applied to recent DNP data. As a simple and practical alternative, Neudert et al. (2017) disentangle the OE and SE components of such mixed DNP spectra using only the integral of the measured cw-EPR signal. Their approach is based on the following insightful observations: (i) up to an overall scaling factor the EPR line shape is equal to the saturation factor and thus to the OE enhancement (Eq. (2)); (ii) up to an overall scaling factor the SE enhancement lines at $\pm\omega_I$ are shifted versions (and flipped for the ZQ transition) of the same EPR line shape. One can thus identify the contributions of OE and SE to the DNP spectrum by placing the integrated cw-EPR spectrum at, respectively, zero and $\pm\omega_I$ offsets, and independently adjusting the magnitudes of the two components.

This approach is illustrated in Figs. 1c and 1d, where the dashed-dotted blue lines are the integrals of the cw-EPR spectra from Figs. 1a and 1b, respectively (flipped here to reflect the dipolar nature of OE), and the dashed green lines are the same EPR spectra but centered at $-400\,\mathrm{MHz}$ and $+400\,\mathrm{MHz}$. The sum of the OE and SE contributions determined in this way is shown with a dotted black line. This sum is seen to agree closely with the DNP spectrum of 10-Doxyl-PC (Fig. 1c) and to capture well the overall shape of the DNP spectrum of 16-Doxyl-PC (Fig. 1d).

In spite of the good general agreement between the experimental DNP spectra and the dotted black lines in Figs. 1c and 1d, some persistent differences remain. In particular, (i) the OE feature in the experiment appears to be consistently broader than the EPR line and (ii) the enhancement between the central OE feature and the negative SE feature is consistently larger than what is predicted by the overlap of the two copies of the EPR line shape. Both of these aspects are especially clear in the case of 16-Doxyl-PC (Fig. 1d). The theory presented below (Sect. 4) aims to address these deficiencies of the simple approach.

In fact, the first deficiency is easy to rationalize. Cw-EPR spectra are recorded at low mw power and their widths reflect mechanisms contributing to the electronic $T_2$ relaxation. The DNP spectrum, on the other hand, is recorded at high mw power, where the EPR line width experiences power-broadening that also depends on the electronic $T_1$ relaxation. That the OE-DNP spectrum "represents an indirect observation of the electron resonance when greatly saturated" was understood early on (Carver and Slichter, 1956, Fig. 6). To properly model the contribution of OE to mixed DNP spectra, therefore, it is necessary to calculate the cw-EPR spectrum under saturating conditions. How to rigorously do that in the regime of slow radical tumbling is known (Freed et al., 1971).

While power-broadening affects OE, it is not immediately clear whether one should also take it into account when modeling SE. (We address this point in Sect. 4.4.) Even leaving power-broadening aside, however, we know that in liquids the SE lines of the DNP spectrum should also be broader than the EPR line width because of the fluctuations of the dipolar interaction (Sezer, 2023b). Although Sezer (2023b) showed how to quantify this additional motional broadening in the case of translational molecular diffusion, the theoretical treatment there assumed a Lorentzian EPR line and is thus not directly applicable to the experiments in Fig. 1. In the current paper, we extend the formalism to slow radical tumbling and g-tensor anisotropy (Sect. 4). In Sect. 5 we apply the developed theory to the analysis of the experimental spectra in Fig. 1, disentangling the contributions of SE and OE to the observed DNP. The needed theoretical background from Sezer (2023a, b) is reviewed next.

## 3  Theoretical background

The classical Bloch equations describe the dynamics of the electronic magnetization, including under saturating conditions. In Sect. 3.1 we recall the relationship between the steady-state solution of the Bloch equations and cw-EPR. Then, in Sect. 3.2, closed-form expressions are obtained for the rate constants of the forbidden transitions that are driven by the microwaves in SE-DNP. These expressions, derived in this form for the first time (Eqs. (33) and (34)), are similar to the steady-state solutions of the Bloch equations but additionally contain (i) the strength of the electron-nucleus dipolar interaction and (ii) the Larmor frequency of the polarized nuclear spin (Sezer, 2023a). Finally, in Sect. 3.3 we remind the reader how these expressions should be modified in the presence of random modulation of the dipolar interaction, as relevant for liquids (Sezer, 2023b).

The reviewed results, which apply to a single Lorentzian line, will be extended in Sect. 4 to the regime of slow radical tumbling and anisotropic g tensor. In the process, some of the scalar variables that appear below, like the offset frequency and the electronic relaxation rates, will be replaced by square matrices, as we explain in Secs. 4.1 and 4.2. The generalization of Secs. 3.1, 3.2 and 3.3 along these lines is carried out in, respectively, Secs. 4.3, 4.4 and 4.5.

### 3.1 Bloch equations

The evolution of the expectation values of the electronic spin operators $S_i$ ($i = x, y, z$), which we denote by $s_i$, is described by the classical Bloch equations (in the rotating frame)

$$
\begin{bmatrix} \dot{s}_x(t) \\ \dot{s}_y(t) \\ \dot{s}_z(t) \end{bmatrix} = - \begin{bmatrix} R_2 & \Delta & 0 \\ -\Delta & R_2 & \omega_1 \\ 0 & -\omega_1 & R_1 \end{bmatrix} \begin{bmatrix} s_x(t) \\ s_y(t) \\ s_z(t) \end{bmatrix} + R_1 \begin{bmatrix} 0 \\ 0 \\ s_z^{\mathrm{eq}} \end{bmatrix}. \tag{8}
$$

Here, the dot above the variable indicates differentiation with respect to time, $R_2$ and $R_1$ are the reciprocals of the electronic relaxation times $T_2$ and $T_1$, respectively, and[3]

$$
\Delta = \omega_0 - \omega \tag{9}
$$

is the offset between the Larmor frequency of the electronic spins, $\omega_0$, and the (angular) frequency of the oscillating magnetic field, $\omega$. In the case of an isotropic g-factor, $g_0$,

$$
\omega_0 = g_0 \mu_{\mathrm{B}} B_0 / \hbar, \tag{10}
$$

where $\mu_{\mathrm{B}}$ is the Bohr magneton.

At steady state

$$
\begin{bmatrix} R_2 & \Delta & 0 \\ -\Delta & R_2 & \omega_1 \\ 0 & -\omega_1 & R_1 \end{bmatrix} \begin{bmatrix} s_x^{\mathrm{ss}} \\ s_y^{\mathrm{ss}} \\ s_z^{\mathrm{ss}} \end{bmatrix} = R_1 \begin{bmatrix} 0 \\ 0 \\ s_z^{\mathrm{eq}} \end{bmatrix}. \tag{11}
$$

Solving these algebraic equations for the variables $s_i^{\mathrm{ss}}$, one can calculate the cw-EPR spectrum and the electronic saturation profile. Making use of the zeros in the first and last rows of the Bloch matrix in Eq. (11), we first express $s_x^{\mathrm{ss}}$ and $s_z^{\mathrm{ss}}$ in terms of $s_y^{\mathrm{ss}}$:

$$
s_x^{\mathrm{ss}} = -\Delta T_2 s_y^{\mathrm{ss}}, \qquad s_z^{\mathrm{ss}} = \omega_1 T_1 s_y^{\mathrm{ss}} + s_z^{\mathrm{eq}}. \tag{12}
$$

The middle row of the matrix then yields

$$
s_y^{\mathrm{ss}} = -\omega_1 P_0^{-1} s_z^{\mathrm{eq}}, \tag{13}
$$

where we defined

$$
P_0 = R_2 + \omega_1^2 T_1 + \Delta^2 T_2. \tag{14}
$$

The in-phase (absorptive) and the out-of-phase (dispersive) components of the cw-EPR signal are then found to be

$$
\mathrm{abs} = s_y^{\mathrm{ss}} / s_z^{\mathrm{eq}} = -\omega_1 P_0^{-1}
$$
$$
\mathrm{dsp} = s_x^{\mathrm{ss}} / s_z^{\mathrm{eq}} = -\Delta T_2 \, \mathrm{abs}. \tag{15}
$$

---

[3]In Sezer (2023a, b) the frequency offset was denoted by $\Omega$. Here we reserve this symbol for the orientation of the polarizing agent (Sect. 4).

From the longitudinal component at steady state we similarly find

$$s = 1 - s_z^{\text{ss}}/s_z^{\text{eq}} = -\omega_1 T_1 \, \text{abs}, \tag{16}$$

which shows that the saturation factor is directly proportional to the absorptive EPR line shape. This proportionality holds for all mw powers, including the large powers used in DNP. In Sect. 4.3 we show that it remains valid also in the case of g-tensor anisotropy.

When generalizing the Bloch equations to an anisotropic g tensor, we will need to work with high-dimensional abstract vectors. To distinguish these vectors from the vectors in 3D space, we will denote the latter by placing an arrow above their symbols, and will use bold symbols for the former. (A 3D unit vector will be indicated with a hat rather than an arrow.) Additionally, we will use capital hollow letters to denote $3 \times 3$ matrices that act on the 3D vectors. With this understanding, we will write the Bloch equations in Eq. (8) as

$$\dot{\vec{s}}(t) = -\mathbb{B}_0 \vec{s}(t) + R_1 \hat{k} s_z^{\text{eq}}, \tag{17}$$

where

$$\vec{s}(t) = \begin{bmatrix} s_x(t) \\ s_y(t) \\ s_z(t) \end{bmatrix}, \qquad \hat{k} = \begin{bmatrix} 0 \\ 0 \\ 1 \end{bmatrix}, \tag{18}$$

$$\mathbb{B} = \begin{bmatrix} R_2 + i\omega_I & \Delta & 0 \\ -\Delta & R_2 + i\omega_I & \omega_1 \\ 0 & -\omega_1 & R_1 + i\omega_I \end{bmatrix}, \tag{19}$$

and $\mathbb{B}_0 = \mathbb{B}(\omega_I = 0)$. The $i\omega_I$ that has been added to the main diagonal of the matrix $\mathbb{B}$ will be needed for the dynamical description of the solid effect (see Sect. 3.2). The subscript of $\mathbb{B}_0$ is intended as a reminder that $\mathbb{B}$ is evaluated at $\omega_I = 0$, where $\omega_I$ is the Larmor frequency of the polarized nuclear spin.

## 3.2 Solid effect in solids

SE relies on the dipolar interaction between the electronic and nuclear spins whose coupling is

$$A_1 = D_{\text{dip}} \frac{-3\cos\theta\sin\theta}{r^3} e^{i\phi}. \tag{20}$$

Here $D_{\text{dip}} = (\mu_0/4\pi)\hbar\gamma_S\gamma_I$ is the dipolar constant, which equals approximately $2\pi(79\,\text{kHz}\,\text{nm}^3)$ for protons, and $(r, \theta, \phi)$ are the spherical polar coordinates of the inter-spin vector.

In liquids, $A_1$ changes in time because of molecular diffusion. The treatment of SE-DNP for time-dependent $A_1$ in Sezer (2023b) was developed under the assumption that the nuclear $T_1$ is orders of magnitude larger than the correlation time of the electron-nucleus dipolar interaction, which is practically always the case in liquids. For the same analysis to apply to

solids, nuclear spin diffusion, which analogously to molecular diffusion in liquids spreads out the nuclear polarization across the sample, should be much faster than the nuclear $T_1$. Although this condition is not necessarily satisfied in the solid state,

for the mathematical description in terms of a dipolar correlation function to apply, we will assume that spin diffusion is fast when referring to solids. Similarly, when accounting for g-tensor anisotropy below, we will assume that the tumbling of the radical is much faster than the nuclear $T_1$. This assumption is clearly violated in solids where "tumbling" is infinitely slow. Nevertheless, for the purposes of comparison, we will refer in the following to 'solids' with the understanding that the correlation time of the dipolar interaction is much shorter than the nuclear $T_1$ (in order to treat nuclear spin diffusion on the

level of a translational correlation function) but much longer than all other relaxation time scales (in order to treat the electron-nucleus dipolar interaction as constant). Because we will keep all other parameters, including the time scale of radical tumbling, the same when comparing 'solids' and liquids, it should be kept in mind that our treatment is not a good model for the solid state (hence the quotation marks).

For SE-DNP, in addition to the Bloch equations it is necessary to consider the following dynamical equations of the electron-

250 nucleus coherences $g_i = \langle S_i I_+ \rangle$ $(i = x, y, z)$ (Sezer, 2023a):

$$
\begin{bmatrix} \dot{g}_x(t) \\ \dot{g}_y(t) \\ \dot{g}_z(t) \end{bmatrix} = -\mathbb{B} \begin{bmatrix} g_x(t) \\ g_y(t) \\ g_z(t) \end{bmatrix} - \frac{1}{4} A_1 \begin{bmatrix} s_y(t) \\ -s_x(t) \\ 0 \end{bmatrix} - \mathrm{i} \frac{1}{4} A_1 \begin{bmatrix} 0 \\ 0 \\ i_z(t) \end{bmatrix}. \tag{21}
$$

Again, we are only interested in the steady state of the dynamics where

$$
\mathbb{B} \begin{bmatrix} g_x^{\mathrm{ss}} \\ g_y^{\mathrm{ss}} \\ g_z^{\mathrm{ss}} \end{bmatrix} = -\frac{1}{4} A_1 \begin{bmatrix} s_y^{\mathrm{ss}} \\ -s_x^{\mathrm{ss}} \\ 0 \end{bmatrix} - \mathrm{i} \frac{1}{4} A_1 \begin{bmatrix} 0 \\ 0 \\ i_z^{\mathrm{ss}} \end{bmatrix}. \tag{22}
$$

The rate constants $pv_-$ and $v_+$ needed to calculate the SE enhancement (Eq. (5)) are determined from $g_z^{\mathrm{ss}}$ using the following

equality, which combines (Sezer, 2023a, Eq. (31)) and (Sezer, 2023b, Eq. (42)):

$$
\dot{i}_z|_{\mathrm{coh}}^{\mathrm{ss}} = -\mathrm{Re}\{\mathrm{i} A_1^* g_z^{\mathrm{ss}}\} = -R_{1I}^A i_z^{\mathrm{ss}} - v_+ i_z^{\mathrm{ss}} - pv_- s_z^{\mathrm{eq}}. \tag{23}
$$

(Re{} takes the real part of its argument.) The term proportional to $R_{1I}^A$ on the right-hand side of Eq. (23) accounts for the contribution of the coherences $g_i$ to the nuclear $T_1$ relaxation in the absence of mw excitation. This contribution should be removed when calculating the mw-related rates $v_+$ and $pv_-$.

To a good approximation the electronic spin dynamics is independent from the dipolar interaction with the nuclear spins, as other mechanisms are more efficient in causing electronic relaxation, especially in liquids. As a result, the steady-state expressions from Sect. 3.1 can be used when solving Eq. (22) for $g_z^{\mathrm{ss}}$.

Inverting the matrix $\mathbb{B}$ in Eq. (22), and using $s_{x,y}^{\mathrm{ss}}$ from before, we find

$$
\begin{aligned}
g_z^{\mathrm{ss}} = {} & \omega_1 \frac{1}{4} A_1 ([\mathbb{B}^{-1}]_{zx} + \Delta T_2 [\mathbb{B}^{-1}]_{zy}) P_0^{-1} s_z^{\mathrm{eq}} \\
& - \mathrm{i} \frac{1}{4} A_1 [\mathbb{B}^{-1}]_{zz} i_z^{\mathrm{ss}},
\end{aligned} \tag{24}
$$

where $[\mathbb{B}^{-1}]_{ij}$ is the $ij$th matrix element of $\mathbb{B}^{-1}$. Substituting this $g_z^{ss}$ into Eq. (23) we identify the desired SE rate constants

$$
\begin{aligned}
R_{1I}^A &= \delta^2 \mathrm{Re}\{[\mathbb{B}_{\omega_1=0}^{-1}]_{zz}\} \\
v_+ &= \delta^2 \mathrm{Re}\{[\mathbb{B}^{-1}]_{zz}\} - R_{1I}^A \\
pv_- &= -\delta^2 \omega_1 P_0^{-1} \mathrm{Im}\{[\mathbb{B}^{-1}]_{zx} + \Delta T_2[\mathbb{B}^{-1}]_{zy}\},
\end{aligned}
\tag{25}
$$

where

$$
\delta^2 = (A_1^* A_1)/4
\tag{26}
$$

reflects the strength of the dipolar interaction. ($\mathrm{Im}\{\}$ takes the imaginary part of its argument.)

In liquids, where $A_1$ is time-dependent, we will need to modify the matrix $\mathbb{B}^{-1}$ in Eq. (25) without changing the structure of these expressions (Sect. 3.3). In the case of solids (i.e., when $A_1$ does not change with time), it is possible to carry out the inversion of $\mathbb{B}$ by expressing $g_x^{ss}$ and $g_z^{ss}$ in terms of $g_y^{ss}$, analogously to our treatment of the Bloch equations in the previous subsection.

    From the upper and lower rows of $\mathbb{B}$ in Eq. (22) we find

$$
\begin{aligned}
g_x^{ss} &= -\Delta (R_2 + i\omega_I)^{-1} g_y^{ss} - \frac{1}{4} A_1 (R_2 + i\omega_I)^{-1} s_y^{ss} \\
\quad g_z^{ss} &= \omega_1 (R_1 + i\omega_I)^{-1} g_y^{ss} - i\frac{1}{4} A_1 (R_1 + i\omega_I)^{-1} i_z^{ss}.
\end{aligned}
\tag{27}
$$

Substituting this $g_z^{ss}$ into Eq. (23) we obtain

$$
\begin{aligned}
\dot{i}_z|_{\mathrm{coh}}^{ss} &= -\omega_1 \mathrm{Re}\{iA_1^* (R_1 + i\omega_I)^{-1} g_y^{ss}\} \\
&\quad - \delta^2 \mathrm{Re}\{(R_1 + i\omega_I)^{-1}\} i_z^{ss}.
\end{aligned}
\tag{28}
$$

The first term on the right-hand side of Eq. (28) vanishes when $\omega_1 = 0$. In contrast, the term in the second line is independent of $\omega_1$ and thus contributes also in the absence of mw excitation. We thus identify this second term with the thermal relaxation
rate

$$
R_{1I}^A = \delta^2 \mathrm{Re}\{(R_1 + i\omega_I)^{-1}\}.
\tag{29}
$$

Since we are not interested in this rate, the second summand in Eq. (28) can be dropped at this stage. The rate constants $v_+$ and $pv_-$ will thus be identified using only the first line in Eq. (28):

$$
\omega_1 \mathrm{Re}\{iA_1^* (R_1 + i\omega_I)^{-1} g_y^{ss}\} = v_+ i_z^{ss} + pv_- s_z^{eq}.
\tag{30}
$$

Substituting $g_x^{ss}$ and $g_z^{ss}$ from Eq. (27) into the middle equality of Eq. (22), and using the electronic steady state, we find

$$
\begin{aligned}
g_y^{ss} &= \frac{1}{4} A_1 \omega_1 \Delta P_0^{-1} [R_2^{-1} + (R_2 + i\omega_I)^{-1}] P^{-1} s_z^{eq} \\
&\quad + i\frac{1}{4} A_1 \omega_1 (R_1 + i\omega_I)^{-1} P^{-1} i_z^{ss},
\end{aligned}
\tag{31}
$$

where

$$P = R_2 + i\omega_I + \omega_1^2(R_1 + i\omega_I)^{-1} + \Delta^2(R_2 + i\omega_I)^{-1} \tag{32}$$

generalizes Eq. (14) such that $P_0 = P(\omega_I = 0)$. Finally, using this $g_y^{ss}$ in Eq. (30) we obtain

$$v_+ = -\delta^2\omega_1^2 \operatorname{Re}\left\{ \frac{(R_1 + i\omega_I)^{-2}}{R_2 + i\omega_I + \frac{\omega_1^2}{R_1 + i\omega_I} + \frac{\Delta^2}{R_2 + i\omega_I}} \right\} \tag{33}$$

and

$$pv_- = -\delta^2\omega_1^2 \frac{\Delta}{R_2 + \omega_1^2 T_1 + \Delta^2 T_2}$$
$$\times \operatorname{Im}\left\{ \frac{[R_2^{-1} + (R_2 + i\omega_I)^{-1}](R_1 + i\omega_I)^{-1}}{R_2 + i\omega_I + \frac{\omega_1^2}{R_1 + i\omega_I} + \frac{\Delta^2}{R_2 + i\omega_I}} \right\}. \tag{34}$$

In these expressions we have written down the combinations $P$ and $P_0$ explicitly in order to show in closed form how $v_+$ and $pv_-$ depend on all parameters. For example, we immediately see that $pv_-$ is odd in the offset $\Delta$ while $v_+$ is even. Because the SE-DNP enhancement is proportional to the ratio of these two rates (Eq. (5)), it has the characteristic odd (i.e., antisymmetric) dependence on the offset from the electronic resonance.

When generalizing the SE spin dynamics to g-tensor anisotropy, we will write the dynamical equations (Eq. (21)) as

$$\dot{\vec{g}}(t) = -\mathbb{B}\vec{g}(t) - \frac{1}{4}A_1\mathbb{G}\vec{s}(t) - i\frac{1}{4}A_1\hat{k}i_z(t) \tag{35}$$

with

$$\vec{g}(t) = \begin{bmatrix} g_x(t) \\ g_y(t) \\ g_z(t) \end{bmatrix}, \qquad \mathbb{G} = \begin{bmatrix} 0 & 1 & 0 \\ -1 & 0 & 0 \\ 0 & 0 & 0 \end{bmatrix} = \frac{\partial\mathbb{B}}{\partial\Delta}. \tag{36}$$

### 3.3 Solid effect in liquids

The modulation of the dipolar interaction by translational diffusion was described in Sezer (2023b) on the level of the spectral density of the motional model, which was denoted by $J_{11}(s)$ since this is the Laplace transform of the auto-correlation function of the dipolar interaction $A_1$ (hence the double subscript of $J$). As an example, the spectral density of the force-free hard-sphere (FFHS) model of translational diffusion is (Ayant et al., 1975; Hwang and Freed, 1975)

$$J_{11}^{ffhs}(s) = \langle\delta^2\rangle\tau \frac{(s\tau)^{\frac{1}{2}} + 4}{(s\tau)^{\frac{3}{2}} + 4(s\tau) + 9(s\tau)^{\frac{1}{2}} + 9}. \tag{37}$$

Here, the parameter

$$\tau = b^2/D_{\text{trans}} \tag{38}$$

is the diffusive time scale of the model, which depends on the contact distance of the electronic and nuclear spins, $b$, and on the coefficient of their relative translational diffusion, $D_{\text{trans}}$, and

$$\langle \delta^2 \rangle = D_{\text{dip}}^2 \frac{6\pi}{5} \frac{N}{3b^3}. \tag{39}$$

is the average of the dipolar interaction strength $\delta^2$ over the sample volume, times the concentration of the electronic spins, $N$.

It is convenient to write $J_{11}$, which has units of angular frequency, as

$$J_{11}(s) = \langle \delta^2 \rangle j_{11}(s), \tag{40}$$

where $j_{11}(s)$ has units of time. This factorization confines the effect of the parameters $N$, $b$ and the constant $D_{\text{dip}}$ to the scaling factor $\langle \delta^2 \rangle$. The factor $j_{11}(s)$ then fully accounts for the line shape of the SE-DNP spectrum, which results from the interplay between the offset frequency and the time scale of the translational motion.

According to Sezer (2023b), the modification from solids to liquids amounts to replacing the matrix $\mathbb{B}^{-1}$ in Eq. (25) by the matrix

$$\mathbb{Q} = j_{11}(\mathbb{B}), \tag{41}$$

and also replacing $\delta^2$ by $\langle \delta^2 \rangle$. The desired SE rate constants in liquids are thus

$$
\begin{aligned}
R_{1I}^A &= \langle \delta^2 \rangle \text{Re}\{[\mathbb{Q}_{\omega_1=0}]_{zz}\} \\
v_+ &= \langle \delta^2 \rangle \text{Re}\{[\mathbb{Q}]_{zz}\} - R_{1I}^A \\
pv_- &= -\langle \delta^2 \rangle \omega_1 P_0^{-1} \text{Im}\{[\mathbb{Q}]_{zx} + \Delta T_2 [\mathbb{Q}]_{zy}\}.
\end{aligned}
\tag{42}
$$

We now clarify the meaning of Eq. (41). Following the definition of a function of a matrix, one should first solve the eigenvalue problem of $\mathbb{B}$, i.e., $\mathbb{B}\mathbb{U} = \mathbb{U}\Lambda$, where the diagonal matrix $\Lambda = \text{diag}(\lambda_1, \lambda_2, \lambda_3)$ contains the three eigenvalues and the columns of $\mathbb{U}$ contain the corresponding (right) eigenvectors. Then one should evaluate the spectral density at the three eigenvalues: $\ell_n = j_{11}(\lambda_n)$. Finally, one should form the diagonal matrix $\mathbb{L} = \text{diag}(\ell_1, \ell_2, \ell_3)$ and calculate $\mathbb{Q} = \mathbb{U}\mathbb{L}\mathbb{U}^{-1}$. Comparing this expression of $\mathbb{Q}$ with $\mathbb{B}^{-1} = \mathbb{U}\Lambda\mathbb{U}^{-1}$ we see that in the transition from solids to liquids, where $\mathbb{B}^{-1}$ is replaced by $\mathbb{Q}$, we essentially "process" the eigenvalues of $\mathbb{B}$ with the spectral density function $j_{11}$. This step prevents us from eliminating the variables $g_{x,z}^{\text{ss}}$ the way we did previously for solids (Sect. 3.2). Because of that, the rate constants in liquids (Eq. (42)) need to be calculated numerically.

Nonetheless, it is still possible to simplify the expression for $R_{1I}^A$ since when $\omega_1 = 0$ the $zz$ component of $\mathbb{B}$ is decoupled from the rest of the matrix. One then finds

$$R_{1I}^A = \langle \delta^2 \rangle \text{Re}\{j_{11}(R_1 + i\omega_I)\}. \tag{43}$$

Clearly, the time-dependence of the dipolar interaction modifies all rate constants, including $R_{1I}^A$ (*cf.* Eq. (29)).

## 4 Slow-motional EPR and DNP spectra for anisotropic g tensor

In this section we show how to account for g-tensor anisotropies when the tumbling of the radical is slow. Because our description of SE is built around the Bloch equations (Sezer, 2023a, b), we first adapt the treatment of isotropic rotational diffusion of Freed et al. (1971) to our needs (Secs. 4.1, 4.2 and 4.3) and then generalize it to SE-DNP (Secs. 4.4 and 4.5). If needed, further generalization to anisotropic diffusion and an orienting potential can be carried out analogously, following the mathematical treatment of the MOMD and SRLS models for slow-motional EPR (Meirovitch et al., 1984; Polimeno and Freed, 1995).

### 4.1 Stochastic Liouville equation for isotropic tumbling

Following Freed et al. (1971), we account for the effect of tumbling on the EPR spectrum using the SLE formalism (Anderson, 1954; Kubo, 1954). We describe the rotational state of the radical statistically with the probability density $P(\Omega, t)$, which quantifies the likelihood that at time $t$ the molecular system of coordinates in which the g tensor is diagonal has orientation $\Omega$ with respect to the laboratory system of axes defined by the magnetic fields $B_0$ and $B_1$. In the case of isotropic rotation this probability evolves with the Fokker-Planck equation

$$\frac{\partial}{\partial t} p(\Omega, t) = D_{\text{rot}} \nabla_\Omega^2 p(\Omega, t), \tag{44}$$

where $D_{\text{rot}}$ is the rotational diffusion constant of the radical and the Laplacian differential operator $\nabla_\Omega^2$ acts on the orientation variable $\Omega$. The operator

$$K_\Omega = -D_{\text{rot}} \nabla_\Omega^2 \tag{45}$$

satisfies the following eigenvalue problem

$$K_\Omega \mathscr{D}_{mn}^\ell(\Omega) = D_{\text{rot}} \ell(\ell+1) \mathscr{D}_{mn}^\ell(\Omega), \tag{46}$$

where the eigenfunctions $\mathscr{D}_{mn}^\ell(\Omega)$ are the Wigner rotation matrix elements, which are orthogonal to each other:

$$\int \mathscr{D}_{MN}^{L*}(\Omega) \mathscr{D}_{mn}^\ell(\Omega) \, d\Omega = \frac{8\pi^2}{2L+1} \delta_{L\ell} \delta_{Mm} \delta_{Nn}. \tag{47}$$

From Eq. (46) it is clear that the time derivative on the left-hand side of Eq. (44) vanishes for the equilibrium probability

$$p^{\text{eq}}(\Omega) = \frac{1}{8\pi^2} = \frac{1}{8\pi^2} \mathscr{D}_{00}^0(\Omega). \tag{48}$$

In the presence of g-tensor anisotropy, the electronic Larmor frequency depends on the orientation $\Omega$ of the radical as follows:[4]

$$\omega(\Omega) = \omega_0 + \gamma_0^2 \mathscr{D}_{00}^2(\Omega) + \gamma_2^2 [\mathscr{D}_{-20}^2(\Omega) + \mathscr{D}_{20}^2(\Omega)], \tag{49}$$

---

[4]We follow Freed et al. (1971) and consider the effect of the g-tensor anisotropy only on the secular terms in the electronic spin Hamiltonian, i.e., those proportional to the spin operator $S_z$. The response of the non-secular terms to the g anisotropy is neglected.

where the angular frequencies (Freed et al., 1971)

$$\gamma_0^2 = \frac{2}{3}\left[g_{zz} - \frac{1}{2}(g_{xx} + g_{yy})\right]\mu_B B_0/\hbar$$

$$\gamma_2^2 = \frac{1}{\sqrt{6}}(g_{xx} - g_{yy})\,\mu_B B_0/\hbar \tag{50}$$

are formed from the components $g_{xx}$, $g_{yy}$ and $g_{zz}$ of the g-tensor in the molecular frame. In Eq. (49), the first index in the subscripts of the Wigner rotation matrix elements refers to molecular system of axes while the second index refers to the laboratory system. The second indices are zero here because we consider only the secular terms, which are proportional to $S_z$.

Since the electronic Larmor frequency depends on $\Omega$, the offset frequency $\Delta$ also becomes a function of the molecular orientation. As an example, for a fixed $\Omega$ the Bloch equations (Eq. (17)) should be modified as

$$\dot{\vec{s}}(t) = -[\mathbb{B}_0 + \mathbb{F}(\Omega)]\vec{s}(t) + R_1\hat{k}s_z^{eq}, \tag{51}$$

where the orientation dependence is confined to the $3 \times 3$ matrix

$$\mathbb{F}(\Omega) = \left\{\gamma_0^2 \mathscr{D}_{00}^2(\Omega) + \gamma_2^2[\mathscr{D}_{-20}^2(\Omega) + \mathscr{D}_{20}^2(\Omega)]\right\}\mathbb{G}. \tag{52}$$

(The matrix $\mathbb{G}$ was introduced in Eq. (36).) It should be stressed, however, that Eq. (51) is not a legitimate equation of motion, as it does not account for the dynamics of the orientation $\Omega$.

The SLE formalism remedies this deficiency by introducing the orientation-conditioned averages $\vec{s}(\Omega, t)$, whose spatial part evolves according to the Bloch equations (Eq. (51)) and whose $\Omega$ dependence evolves according to the diffusion equation (Eq. (44)):

$$\frac{\partial}{\partial t}\vec{s}(\Omega, t) = -(K_\Omega \otimes \mathbb{E} + E_\Omega \otimes \mathbb{B}_0)\,\vec{s}(\Omega, t)$$

$$- E_\Omega \otimes \mathbb{F}(\Omega)\,\vec{s}(\Omega, t) + R_1\hat{k}s_z^{eq}p^{eq}(\Omega). \tag{53}$$

Here $\mathbb{E}$ is the $3 \times 3$ identity matrix in 3D space and $E_\Omega$ is the identity operator in the same abstract space as $K_\Omega$. The outer product $\otimes$ is needed to create a combined operator that acts simultaneously in both of these spaces.

Since the functions $\mathscr{D}_{mn}^\ell(\Omega)$ form a complete set, we expand $\vec{s}(\Omega, t)$ as follows:

$$\vec{s}(\Omega, t) = \frac{1}{8\pi^2}\sum_{\ell=0}^\infty \sum_{m=-\ell}^\ell \sum_{n=-\ell}^\ell \mathscr{D}_{mn}^\ell(\Omega)\,\vec{s}_{mn}^\ell(t). \tag{54}$$

The coefficients $\vec{s}_{mn}^\ell$, which contain the time-dependence, can be obtained from $\vec{s}(\Omega, t)$ using the orthogonality of $\mathscr{D}_{mn}^\ell(\Omega)$ (Eq. (47)):

$$\vec{s}_{MN}^L(t) = (2L+1)\int \mathscr{D}_{MN}^{L*}(\Omega)\,\vec{s}(\Omega, t)\,d\Omega. \tag{55}$$

Ultimately, the only property that we care about is the integral of the SLE variable $\vec{s}(\Omega, t)$ over all orientations:

$$\int \vec{s}(\Omega, t)\,d\Omega = \int \mathscr{D}_{00}^0(\Omega)\vec{s}(\Omega, t)\,d\Omega = \vec{s}_{00}^0(t). \tag{56}$$

In that sense, the (vector) coefficient $\vec{s}_{00}^{\,0}(t)$ is the main object of interest, while all other coefficients $\vec{s}_{mn}^{\,\ell}(t)$ play an auxiliary, book-keeping role.

Substituting $\vec{s}(\Omega, t)$ from Eq. (54) into Eq. (53), multiplying both sides by $\mathscr{D}_{MN}^{L*}(\Omega)$ and integrating over $\Omega$, we get

$$\dot{\vec{s}}_{MN}^{\,L}(t) = R_1 \hat{k} s_z^{\mathrm{eq}} \delta_{L0} \delta_{M0} \delta_{N0}$$
$$- [D_{\mathrm{rot}} L(L+1) + \mathbb{B}_0] \vec{s}_{MN}^{\,L}(t)$$
$$- \sum_{\ell m n} \left( \frac{2L+1}{8\pi^2} \int \mathscr{D}_{MN}^{L*}(\Omega) \mathscr{D}_{mn}^{\ell}(\Omega) \mathbb{F}(\Omega) \, \mathrm{d}\Omega \right) \vec{s}_{mn}^{\,\ell}(t). \tag{57}$$

Clearly, the terms proportional to $K_\Omega$ and $\mathbb{B}_0$ in Eq. (53) do not mix coefficients $\vec{s}_{MN}^{\,L}$ with different values of $L$, $M$ and $N$. In other words, these two operators are diagonal in the selected representation. The term proportional to $\mathbb{F}(\Omega)$, on the other hand, mixes coefficients with different $L$ and $M$ (but not $N$, as we discuss below).

The integral in the last line of Eq. (57) contains the product of three Wigner rotation matrix elements. These can be expressed in therms of the Clebsch-Gordan coefficients $C_{\ell_1 m_1 \ell_2 m_2}^{LM}$. Specifically, for the $\mathscr{D}_{K0}^{2}(\Omega)$ in Eq. (52), we have

$$\frac{2L+1}{8\pi^2} \int \mathscr{D}_{MN}^{L*}(\Omega) \mathscr{D}_{K0}^{2}(\Omega) \mathscr{D}_{mn}^{\ell}(\Omega) \, \mathrm{d}\Omega = C_{2K\ell m}^{LM} C_{20\ell n}^{LN}, \tag{58}$$

which leads to

$$\dot{\vec{s}}_{MN}^{\,L}(t) = R_1 \hat{k} s_z^{\mathrm{eq}} \delta_{L0} \delta_{M0} \delta_{N0}$$
$$- [D_{\mathrm{rot}} L(L+1) + \mathbb{B}_0] \vec{s}_{MN}^{\,L}(t)$$
$$- \sum_{\ell m n} [\gamma_0^2 C_{20\ell m}^{LM} + \gamma_2^2 (C_{2-2\ell m}^{LM} + C_{22\ell m}^{LM})] C_{20\ell n}^{LN} \mathbb{G} \vec{s}_{mn}^{\,\ell}(t). \tag{59}$$

In Eq. (59), the sum over $\ell$ mixes only expansion coefficients with $\ell = L, L \pm 2$ (Freed et al., 1971) because all three Wigner rotation matrix elements in $\mathbb{F}$ have $L = 2$ (Eq. (52)). Since we need $\vec{s}_{00}^{\,0}$ at the end, it is sufficient to consider only coefficients with even values of $\ell$. Furthermore, as the Wigner rotation matrix elements in $\mathbb{F}$ have $M = 0, \pm 2$ and $N = 0$, the sum over $m$ mixes only coefficients whose values $m$ are either equal to $M$ or differ from it by two units, while the sum over $n$ does not mix any coefficients with $n$ different from $N$. These considerations imply that the triple sum in Eq. (59) will go only over $\vec{s}_{m0}^{\,\ell}$ with even $\ell$ and $m$. Finally, because the Wigner rotation matrix elements with $M = 2$ and $M = -2$ appear in a symmetrical way in $\mathbb{F}$, it becomes possible to work with the symmetrized coefficients (Freed et al., 1971)

$$\vec{s}^{\,LM} = \frac{1}{2}(\vec{s}_{-M0}^{\,L} + \vec{s}_{M0}^{\,L}), \tag{60}$$

thus restricting $M$ to non-negative values ($0 \leqslant M \leqslant L$). The lowest-order coefficients that are coupled by the SLE dynamics are thus $\vec{s}^{\,00}$, $\vec{s}^{\,20}$, $\vec{s}^{\,22}$, $\vec{s}^{\,40}$, $\vec{s}^{\,42}$, $\vec{s}^{\,44}$, $\vec{s}^{\,60}$, etc.

## 4.2 Matrix representation of the SLE dynamics

While the above considerations greatly reduce the needed coefficients, there is still an infinite number left. In any practical work, this infinite set is truncated by selecting a maximum value of $L$ to account for, and setting to zero the coefficients

with $L > L_{max}$. Since the total number of even $L$ such that $L \leqslant L_{max}$ is $n_L = L_{max}/2 + 1$, the total number of remaining coefficients $\vec{s}^{LM}$ is $n_{tot} = n_L(n_L + 1)/2 = L_{max}^2/8 + 3L_{max}/4 + 1$. For the smallest non-trivial choice of $L_{max} = 2$, $n_{tot} = 3$ (with $\vec{s}^{00}$, $\vec{s}^{20}$ and $\vec{s}^{22}$). The number of coefficients increases quadratically with $L_{max}$ (e.g., $n_{tot} = 15, 28, 45$ for $L_{max} = 8, 12, 16$, respectively).

To compactly write down how these coefficients are mixed by the SLE dynamics, we introduce the following abstract vectors with $n_{tot}$ elements:

$$
\mathbf{1}^{00} = \begin{bmatrix} 1 \\ 0 \\ 0 \\ \vdots \end{bmatrix}, \qquad \boldsymbol{s}_i(t) = \begin{bmatrix} s_i^{00}(t) \\ s_i^{20}(t) \\ s_i^{22}(t) \\ \vdots \end{bmatrix} \quad (i = x, y, z), \tag{61}
$$

where the former is needed for the first term on the right-hand side of Eq. (59). The SLE dynamics then becomes

$$
\begin{bmatrix} \dot{\boldsymbol{s}}_x(t) \\ \dot{\boldsymbol{s}}_y(t) \\ \dot{\boldsymbol{s}}_z(t) \end{bmatrix} = -\mathcal{B}_0 \begin{bmatrix} \boldsymbol{s}_x(t) \\ \boldsymbol{s}_y(t) \\ \boldsymbol{s}_z(t) \end{bmatrix} + R_1 \begin{bmatrix} 0 \\ 0 \\ \mathbf{1}^{00} \end{bmatrix} s_z^{eq}, \tag{62}
$$

where

$$
\mathcal{B}_0 = \begin{bmatrix} \mathbf{R}_2 & \boldsymbol{\Delta} & 0 \\ -\boldsymbol{\Delta} & \mathbf{R}_2 & \omega_1 \mathbf{E} \\ 0 & -\omega_1 \mathbf{E} & \mathbf{R}_1 \end{bmatrix} \tag{63}
$$

is a $3n_{tot} \times 3n_{tot}$ matrix, and $\mathbf{E}$, $\mathbf{R}_1$, $\mathbf{R}_2$ and $\boldsymbol{\Delta}$ are $n_{tot} \times n_{tot}$ matrices.

The first three of these sub-matrices are purely diagonal: $\mathbf{E}$ is the identity matrix and

$$
\mathbf{R}_{1,2} = R_{1,2}\mathbf{E} + D_{rot}\mathbf{C}_D, \tag{64}
$$

with the diagonal elements of $\mathbf{C}_D$ being equal to $L(L+1)$. For the simplest case of $L_{max} = 2$ with only three coefficients ($\vec{s}^{00}$, $\vec{s}^{20}$ and $\vec{s}^{22}$),

$$
\mathbf{E} = \begin{bmatrix} 1 & & \\ & 1 & \\ & & 1 \end{bmatrix}, \qquad \mathbf{C}_D = \begin{bmatrix} 0 & & \\ & 6 & \\ & & 6 \end{bmatrix}. \tag{65}
$$

In Eq. (63), the diagonal matrices $\mathbf{R}_{1,2}$ and $\mathbf{E}$, which originate from the second line of Eq. (59), do not mix coefficients with different $L$ and $M$. Only the sub-matrix $\boldsymbol{\Delta}$, which is of the form

$$
\boldsymbol{\Delta} = \Delta\mathbf{E} + \gamma_0^2\mathbf{C}_0 + \gamma_2^2\mathbf{C}_2, \tag{66}
$$

mixes coefficients of different orders. In fact, the mixing is due to the matrices $\mathbf{C}_{0,2}$, which modify the frequency offset $\Delta$ in proportion to the g-tensor anisotropies $\gamma_0^2$ and $\gamma_2^2$. For $L_{\max} = 2$,

$$
\mathbf{C}_0 = \begin{bmatrix} 0 & \frac{1}{5} & 0 \\ 1 & \frac{2}{7} & 0 \\ 0 & 0 & -\frac{2}{7} \end{bmatrix}, \qquad \mathbf{C}_2 = \begin{bmatrix} 0 & 0 & \frac{1}{5} \times 2 \\ 0 & 0 & -\frac{2}{7} \times 2 \\ 1 & -\frac{2}{7} & 0 \end{bmatrix}. \tag{67}
$$

(The factors of two in the last column of $\mathbf{C}_2$ arise from the fact that coefficients with $M = 0$ pose an exception to the sym-metrization in Eq. (60).) The matrix elements of these two matrices in the most general case are

$$
\begin{aligned}
[\mathbf{C}_0]_{LM,\ell m} &= C_{20\ell m}^{LM} C_{20\ell 0}^{L0} \\
[\mathbf{C}_2]_{LM,\ell m} &= (C_{2-2\ell m}^{LM} + C_{22\ell m}^{LM} + \delta_{M0} C_{22\ell m}^{LM}) C_{20\ell 0}^{L0},
\end{aligned} \tag{68}
$$

where the summand proportional to $\delta_{M0}$ in the second line accounts for the factor of two that is needed by the coefficients $\vec{s}^{L0}$.

Selecting $L_{\max} = 0$ in the above formalism amounts to retaining only the (3D vector) coefficient $\vec{s}^{00}$. Then the matrix $\mathcal{B}_0$ in Eq. (63) reduces to $\mathbb{B}_0$, and Eq. (62) reduces to the classical Bloch equations for a homogeneous line. For $L_{\max} > 0$, the diagonal matrices $\mathbf{R}_1$ and $\mathbf{R}_2$ cause the coefficients $s_z^{LM}$ and $s_{x,y}^{LM}$, respectively, to decay exponentially, with those with larger $L$ being suppressed more strongly by the tumbling. Analogously to the Bloch equations, the mw excitation mixes the $y$ and $z$ components of $\vec{s}^{LM}$, without mixing their $LM$ dependence. The latter is mixed only by the offset matrix $\boldsymbol{\Delta}$, as elaborated above.

By building the SLE dynamics on top of the classical Bloch equations, we have arrived at a rather intuitive picture of how the g-tensor anisotropy is incorporated into the spin dynamics. Specifically, every element of the Bloch matrix $\mathbb{B}_0$ (Eq. (19) with $\omega_I = 0$) is replaced by a matrix in the space of $LM$ indices (Eq. (63)). In this replacement, all elements except the frequency offset become diagonal matrices in the $LM$ space, with the mixing in this space being entirely due to the offset. Since we describe the solid effect by two coupled Bloch equations, this intuition about the effect of g-tensor anisotropy on the spin dynamics will be helpful when adapting the approach to SE-DNP.

## 4.3 EPR spectrum and saturation

The cw-EPR spectrum and the electronic saturation factor under g-tensor anisotropy are obtained from the steady state of Eq. (62),

$$
\mathcal{B}_0 \begin{bmatrix} \boldsymbol{s}_x^{\mathrm{ss}} \\ \boldsymbol{s}_y^{\mathrm{ss}} \\ \boldsymbol{s}_z^{\mathrm{ss}} \end{bmatrix} = R_1 \begin{bmatrix} 0 \\ 0 \\ \mathbf{1}^{00} \end{bmatrix} s_z^{\mathrm{eq}}, \tag{69}
$$

which can be solved by inverting the $3n_{\mathrm{tot}} \times 3n_{\mathrm{tot}}$ matrix $\mathcal{B}_0$ numerically. However, it is also possible to solve Eq. (69) by inverting a single matrix with dimensions that are three times smaller (i.e., $n_{\mathrm{tot}} \times n_{\mathrm{tot}}$), as we show next.

First, taking advantage of the zeros in $\mathcal{B}_0$ (Eq. (63)), we express $s_x^{\text{ss}}$ and $s_z^{\text{ss}}$ in terms of $s_y^{\text{ss}}$:

$$s_x^{\text{ss}} = -\mathbf{R}_2^{-1}\boldsymbol{\Delta}\, s_y^{\text{ss}}$$
$$s_z^{\text{ss}} = \omega_1\mathbf{R}_1^{-1}s_y^{\text{ss}} + R_1\mathbf{R}_1^{-1}\mathbf{1}^{00}s_z^{\text{eq}}. \tag{70}$$

Because only the first element of $\mathbf{1}^{00}$ is non-zero, and the diagonal matrix $\mathbf{R}_1$ does not mix coefficients with different values of $LM$, the second equality in Eq. (70) becomes

$$\quad s_z^{\text{ss}} = \omega_1\mathbf{R}_1^{-1}s_y^{\text{ss}} + \mathbf{1}^{00}s_z^{\text{eq}}. \tag{71}$$

For the 00th (i.e., first) element of $s_z^{\text{ss}}$ we thus have $s_z^{00} = \omega_1 T_1 s_y^{00} + s_z^{\text{eq}}$, which is identical to the second equality in Eq. (12). We thus conclude that the proportionality between the electronic saturation factor and the in-phase EPR line shape (Eq. (16)) is not limited to a homogenous line but applies also under g-tensor anisotropy, at least in the case of isotropic rotational diffusion.

Second, from the middle row of the matrix $\mathcal{B}_0$ (Eq. (63)), and after substituting $s_x^{\text{ss}}$ and $s_z^{\text{ss}}$ from Eq. (70), we find

$$\quad s_y^{\text{ss}} = -\omega_1\mathbf{P}_0^{-1}\mathbf{1}^{00}s_z^{\text{eq}}, \tag{72}$$

where we have introduced the $n_{\text{tot}} \times n_{\text{tot}}$ matrix

$$\mathbf{P} = (\mathbf{R}_2 + \mathrm{i}\omega_I) + \omega_1^2(\mathbf{R}_1 + \mathrm{i}\omega_I)^{-1} + \boldsymbol{\Delta}(\mathbf{R}_2 + \mathrm{i}\omega_I)^{-1}\boldsymbol{\Delta} \tag{73}$$

and $\mathbf{P}_0 = \mathbf{P}(\omega_I = 0)$. The matrix $\mathbf{P}_0$ generalizes $P_0$ (Eq. (14)) and Eq. (72) generalizes Eq. (13) to the case of g-tensor anisotropy.

From the 00th components of $s_y^{\text{ss}}$ and $s_x^{\text{ss}}$ we find

$$\text{abs} = -\omega_1[\mathbf{P}_0^{-1}]_{11}, \qquad \text{dsp} = \omega_1 T_2[\boldsymbol{\Delta}\mathbf{P}_0^{-1}]_{11}, \tag{74}$$

where we used the fact that $\mathbf{R}_2$ is a diagonal matrix. These expressions generalize Eq. (15) to the case of g-tensor anisotropy. The corresponding saturation factor as a function of the offset is then (from Eqs. (16) and (74))

$$s(\Delta) = \omega_1^2 T_1[\mathbf{P}_0^{-1}(\Delta)]_{11}. \tag{75}$$

As claimed, to solve for the steady state numerically we need to invert the matrix $\mathbf{P}_0$ whose dimensions are three times smaller than those of $\mathcal{B}_0$. (The two matrix inversions needed to calculate $\mathbf{P}_0$ itself involve the diagonal matrices $\mathbf{R}_{1,2}$.)

The cw-EPR spectrum in derivative mode can be calculated from the derivative of $\mathbf{P}_0$ with respect to the (scalar) frequency offset $\Delta$:

$$\frac{\partial\mathbf{P}_0}{\partial\Delta} = \mathbf{R}_2^{-1}\boldsymbol{\Delta} + \boldsymbol{\Delta}\mathbf{R}_2^{-1}. \tag{76}$$

The in-phase and out-of-phase derivative spectra are then obtained from the first (i.e., 00th) components of the vectors

$$\frac{\partial s_y}{\partial\Delta} = \omega_1\mathbf{P}_0^{-1}(\mathbf{R}_2^{-1}\boldsymbol{\Delta} + \boldsymbol{\Delta}\mathbf{R}_2^{-1})\mathbf{P}_0^{-1}\mathbf{1}^{00}s_z^{\text{eq}}$$
$$\frac{\partial s_x}{\partial\Delta} = -\mathbf{R}_2^{-1}\left(s_y + \boldsymbol{\Delta}\frac{\partial s_y}{\partial\Delta}\right). \tag{77}$$

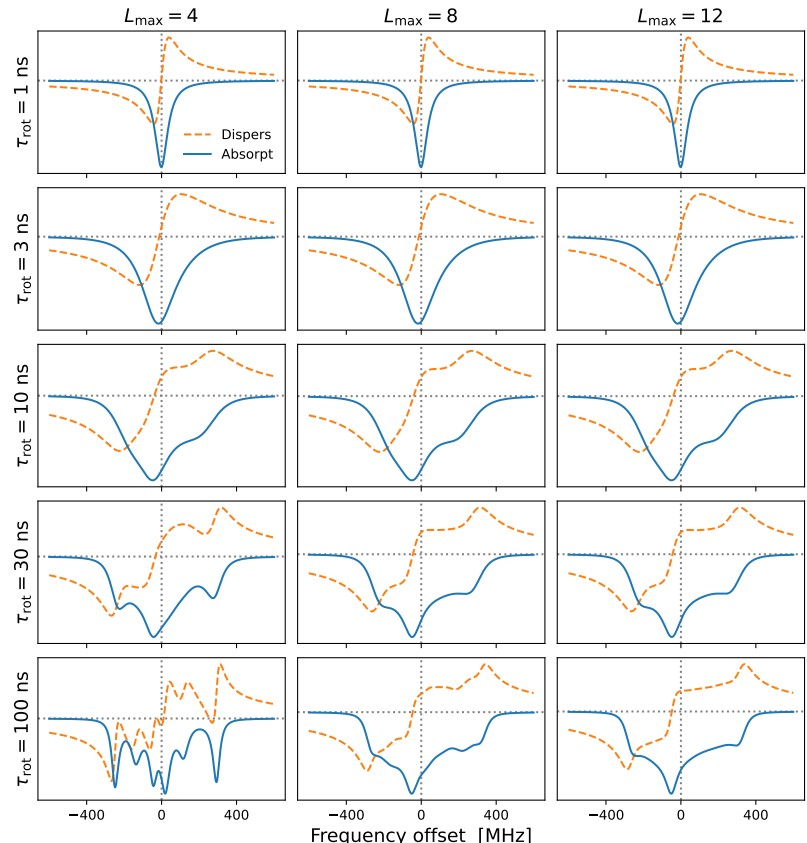

**Figure 2.** Continuous-wave EPR spectra for $g = \mathrm{diag}(2.00755, 2.00555, 2.0023)$ and different tumbling times $\tau_{\mathrm{rot}}$. Larger $L_{\mathrm{max}}$ is necessary for slower tumbling. The needed $L_{\mathrm{max}}$ also depends on the anisotropies, which are $(\gamma_0, \gamma_2) = (-373, 107)$ MHz for $B_0 = 9.403$ T. Other simulation parameters were $B_1 = 0.02$ G, $T_1 = 100$ ns, $T_2^{\mathrm{homog}} = 20$ ns.

These expressions are used in Sect. 5 to fit the experimental EPR spectra from Fig. 1.

In Fig. 2 we show examples of (integral) EPR spectra calculated using the presented approach for different tumbling times $\tau_{\mathrm{rot}}$. The different columns in the figure correspond to different choices of $L_{\mathrm{max}}$. The g-tensor values used in the simulations are characteristic of nitroxide spin labels. We also selected a small mw magnetic field ($B_1 = 0.02$ G) to mimic the low-power conditions typical for cw-EPR. The main message of this figure is that slower tumbling requires larger $L_{\mathrm{max}}$. At the same time, we see that $L_{\mathrm{max}} = 8$ is already good enough for $\tau_{\mathrm{rot}} \leqslant 10$ ns, which is the range of rotational time scales of relevance to our experimental data (Sect. 5). By selecting $L_{\mathrm{max}} = 10$, to be on the safe side, we only need to invert a $21 \times 21$ matrix at every frequency offset, which makes the calculation of g-broadened EPR spectra very fast. This allows us to perform an automated search over the various parameters and fit the experimental cw-EPR spectra in less than a minute.

## 4.4 Solid effect in 'solids'

Extending the above treatment to SE-DNP, we combine the spin dynamics in Eq. (35) with the rotational dynamics in Eq. (44) to form the following SLE:

$$
\frac{\partial}{\partial t}\vec{g}(\Omega, t) = -(K_\Omega \otimes \mathbb{E} + E_\Omega \otimes \mathbb{B})\vec{g}(\Omega, t)
$$
$$
- E_\Omega \otimes \mathbb{F}(\Omega)\vec{g}(\Omega, t)
$$
$$
- \frac{1}{4}A_1 \mathbb{G}\vec{s}(\Omega, t) - \mathrm{i}\frac{1}{4}A_1 \hat{k}i_z(t)p^{\mathrm{eq}}(\Omega). \tag{78}
$$

As before, we introduce the expansion

$$
\vec{g}(\Omega, t) = \frac{1}{8\pi^2} \sum_{\ell=0}^{\infty} \sum_{m=-\ell}^{\ell} \mathscr{D}_{m0}^2(\Omega)\vec{g}_{m0}^\ell(t), \tag{79}
$$

where we have set $n = 0$ from the start, and find

$$
\dot{\vec{g}}_{M0}^L(t) = -[D_{\mathrm{rot}}L(L+1) + \mathbb{B}]\vec{g}_{M0}^L(t)
$$
$$
- \sum_{\ell m}[\gamma_0^2 C_{20\ell m}^{LM} + \gamma_2^2(C_{2-2\ell m}^{LM} + C_{22\ell m}^{LM})]C_{20\ell n}^{L0}\mathbb{G}\vec{g}_{m0}^\ell(t)
$$
$$
- \frac{1}{4}A_1 \mathbb{G}\vec{s}_{M0}^L(t) - \mathrm{i}\frac{1}{4}A_1 \hat{k}\, i_z(t)\delta_{L0}\delta_{M0}. \tag{80}
$$

Again, we switch to the symmetrized coefficients

$$
\vec{g}^{LM} = \frac{1}{2}(\vec{g}_{-M0}^L + \vec{g}_{M0}^L) \tag{81}
$$

and form the following three, $n_{\mathrm{tot}}$-dimensional vectors from the spatial components of the 3D vectors $\vec{g}^{LM}$:

$$
\boldsymbol{g}_i = \begin{bmatrix} g_i^{00} \\ g_i^{20} \\ g_i^{22} \\ \vdots \end{bmatrix} \qquad (i = x, y, z). \tag{82}
$$

The steady state of the resulting spin dynamics is then

$$
\mathcal{B}\begin{bmatrix} \boldsymbol{g}_x^{\mathrm{ss}} \\ \boldsymbol{g}_y^{\mathrm{ss}} \\ \boldsymbol{g}_z^{\mathrm{ss}} \end{bmatrix} = -\frac{1}{4}A_1 \begin{bmatrix} \boldsymbol{s}_y^{\mathrm{ss}} \\ -\boldsymbol{s}_x^{\mathrm{ss}} \\ 0 \end{bmatrix} - \mathrm{i}\frac{1}{4}A_1 \begin{bmatrix} 0 \\ 0 \\ \boldsymbol{1}^{00} \end{bmatrix} i_z^{\mathrm{ss}}, \tag{83}
$$

where

$$
\mathcal{B} = \mathcal{B}_0 + \mathrm{i}\omega_I \tag{84}
$$

generalizes the matrix $\mathcal{B}_0$ from Eq. (63).

Our goal is to solve for $\boldsymbol{g}_z^{\text{ss}}$, since its 00th component should be used in Eq. (23) to calculate the rate constants $pv_-$ and $v_+$. After inverting $\mathcal{B}$ in Eq. (83) we find

$$
\begin{aligned}
\boldsymbol{g}_z^{\text{ss}} = &-\frac{1}{4}A_1[\mathcal{B}^{-1}]_{zx}\boldsymbol{s}_y^{\text{ss}} - \frac{1}{4}A_1[\mathcal{B}^{-1}]_{zy}(-\boldsymbol{s}_x^{\text{ss}}) \\
&- \mathrm{i}\frac{1}{4}A_1[\mathcal{B}^{-1}]_{zz}\mathbf{1}^{00}i_z^{\text{ss}}.
\end{aligned}
\tag{85}
$$

Note that now $[\mathcal{B}^{-1}]_{ij}$ denotes the $n_{\text{tot}} \times n_{\text{tot}}$ sub-matrix of $\mathcal{B}^{-1}$ at position $ij$, and not a scalar matrix element. Using $\boldsymbol{s}_{x,y}^{\text{ss}}$ from the previous subsection, we find that the first component of $\boldsymbol{g}_z^{\text{ss}}$ is

$$
\begin{aligned}
g_z^{00} = &\omega_1 \frac{1}{4}A_1[([\mathcal{B}^{-1}]_{zx} + [\mathcal{B}^{-1}]_{zy}\mathbf{R}_2^{-1}\boldsymbol{\Delta})\mathbf{P}_0^{-1}]_{11}s_z^{\text{eq}} \\
&- \mathrm{i}\frac{1}{4}A_1[[\mathcal{B}^{-1}]_{zz}]_{11}i_z^{\text{ss}}.
\end{aligned}
\tag{86}
$$

Substituting this result into Eq. (23) we obtain

$$
R_{1I}^A = \delta^2\mathrm{Re}\{[[\mathcal{B}_{\omega_1=0}^{-1}]_{zz}]_{11}\}
$$

$$
v_+ = \delta^2\mathrm{Re}\{[[\mathcal{B}^{-1}]_{zz}]_{11}\} - R_{1I}^A
$$

$$
pv_- = -\delta^2\omega_1\mathrm{Im}\{[([\mathcal{B}^{-1}]_{zx} + [\mathcal{B}^{-1}]_{zy}\mathbf{R}_2^{-1}\boldsymbol{\Delta})\mathbf{P}_0^{-1}]_{11}\}.
\tag{87}
$$

These expressions, which require the inversion of the $3n_{\text{tot}} \times 3n_{\text{tot}}$ matrix $\mathcal{B}$, are directly generalizable to liquids (Sect. 4.5). In 'solids', it is possible to obtain alternative expressions that require the inversion of a smaller, $n_{\text{tot}} \times n_{\text{tot}}$ matrix. To this end, we express $\boldsymbol{g}_x^{\text{ss}}$ and $\boldsymbol{g}_z^{\text{ss}}$ in terms of $\boldsymbol{g}_y^{\text{ss}}$ using the first and last rows of $\mathcal{B}$:

$$
\boldsymbol{g}_x^{\text{ss}} = -(\mathbf{R}_2 + \mathrm{i}\omega_I)^{-1}\boldsymbol{\Delta}\boldsymbol{g}_y^{\text{ss}} - \frac{1}{4}A_1(\mathbf{R}_2 + \mathrm{i}\omega_I)^{-1}\boldsymbol{s}_y^{\text{ss}}
$$

$$
\boldsymbol{g}_z^{\text{ss}} = \omega_1(\boldsymbol{R}_1 + \mathrm{i}\omega_I)^{-1}\boldsymbol{g}_y^{\text{ss}} - \mathrm{i}\frac{1}{4}A_1(\boldsymbol{R}_1 + \mathrm{i}\omega_I)^{-1}\mathbf{1}^{00}i_z^{\text{ss}}.
\tag{88}
$$

Substituting the first (i.e., 00th) component of $\boldsymbol{g}_z^{\text{ss}}$ in Eq. (23) we find

$$
R_{1I}^A = \delta^2\mathrm{Re}\{[(\boldsymbol{R}_1 + \mathrm{i}\omega_I)^{-1}]_{11}\}.
\tag{89}
$$

Because $\mathbf{R}_1$ is a diagonal matrix, this result is identical to Eq. (29), showing that $R_{1I}^A$ is not affected by the anisotropy of the g tensor.

Similarly, from the middle part of $\mathcal{B}$ we obtain

$$
\begin{aligned}
\mathbf{P}\boldsymbol{g}_y^{\text{ss}} = &\frac{1}{4}A_1[\boldsymbol{s}_x^{\text{ss}} - \boldsymbol{\Delta}(\mathbf{R}_2 + \mathrm{i}\omega_I)^{-1}\boldsymbol{s}_y^{\text{ss}}] \\
&+ \omega_1\mathrm{i}\frac{1}{4}A_1(\boldsymbol{R}_1 + \mathrm{i}\omega_I)^{-1}\mathbf{1}^{00}i_z^{\text{ss}}.
\end{aligned}
\tag{90}
$$

We first observe that $(\mathbf{R}_1 + \mathrm{i}\omega_I)^{-1}\mathbf{1}^{00} = \mathbf{1}^{00}(R_1 + \mathrm{i}\omega_I)^{-1}$ because $\mathbf{R}_1$ is diagonal. Then we substitute $\boldsymbol{s}_{x,y}^{\text{ss}}$ from before to get

$$
\begin{aligned}
\boldsymbol{g}_y^{\text{ss}} = &\omega_1\frac{1}{4}A_1\mathbf{P}^{-1}[\mathbf{R}_2^{-1}\boldsymbol{\Delta} + \boldsymbol{\Delta}(\mathbf{R}_2 + \mathrm{i}\omega_I)^{-1}]\mathbf{P}_0^{-1}\mathbf{1}^{00}s_z^{\text{eq}} \\
&+ \omega_1\mathrm{i}\frac{1}{4}A_1\mathbf{P}^{-1}\mathbf{1}^{00}(R_1 + \mathrm{i}\omega_I)^{-1}i_z^{\text{ss}}.
\end{aligned}
\tag{91}
$$

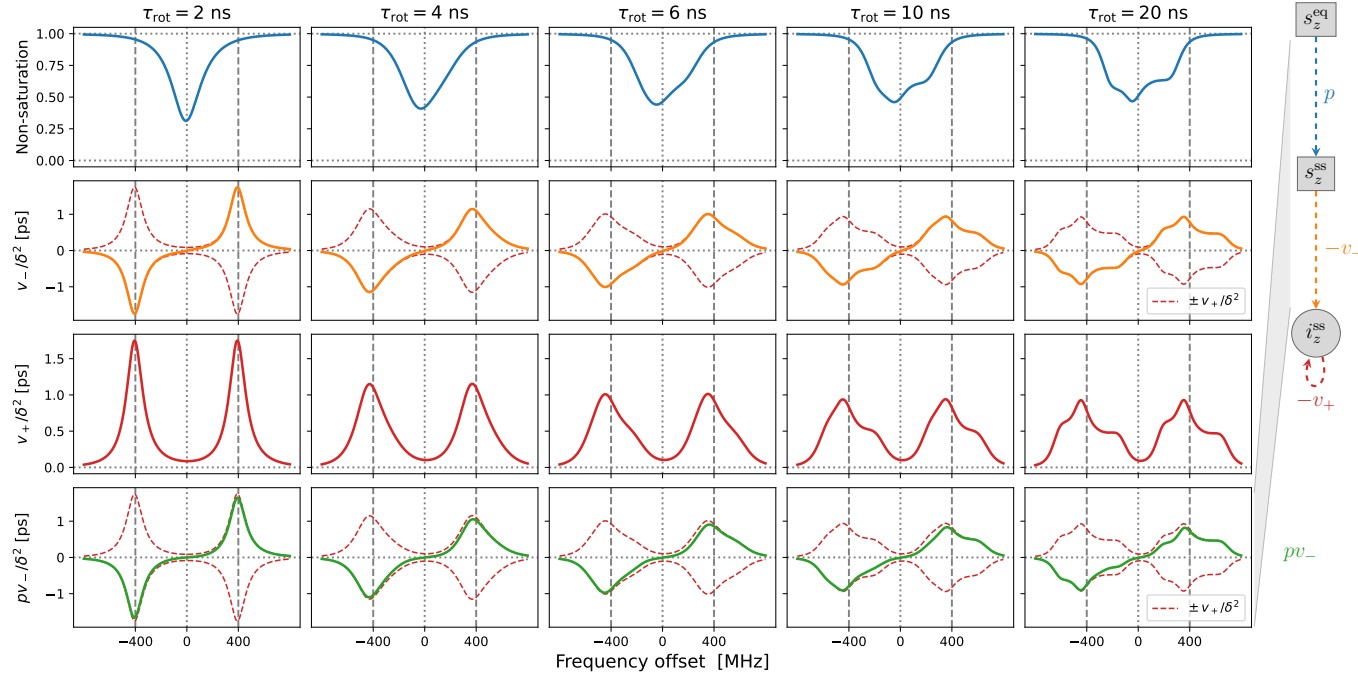

**Figure 3.** Solid-effect rates $v_+$ (red line) and $pv_-$ (green line) calculated at high mw power ($B_1 = 5.5\,\text{G}$) for static dipolar interaction (i.e., 'solid') and several different rates of rotational tumbling. The factorization of $pv_-$ into $p$ (blue line) and $v_-$ (orange line) is also shown. All other parameters as in Fig. 2 and $L_{\max} = 10$. In particular, $T_1 = 100\,\text{ns}$.

Finally, substituting the 00th element of $\boldsymbol{g}_y^{\text{ss}}$ in Eq. (30) we find

$$v_+ = -\delta^2\omega_1^2\,\text{Re}\{(R_1 + i\omega_I)^{-2}[\mathbf{P}^{-1}]_{11}\}$$

$$pv_- = -\delta^2\omega_1^2\text{Im}\{(R_1 + i\omega_I)^{-1}$$

$$\times\,[\mathbf{P}^{-1}(\mathbf{R}_2^{-1}\boldsymbol{\Delta} + \boldsymbol{\Delta}(\mathbf{R}_2 + i\omega_I)^{-1})\mathbf{P}_0^{-1}]_{11}\}. \tag{92}$$

Observe how these expressions generalize Eqs. (33) and (34) to the case of g-tensor anisotropy.

In the last two rows of Fig. 3 we show $v_+/\delta^2$ and $pv_-/\delta^2$, which have units of time. Although the electronic non-saturation factor $p$ and the rate constant $v_-$ always appear together as $pv_-$, it is helpful to separate these two factors when rationalizing SE. We show $p$ and $v_-/\delta^2$ in the first two rows of Fig. 3. Note that $v_+/\delta^2$ and $pv_-/\delta^2$ were calculated directly from Eq. (92), whereas $v_-/\delta^2$ was determined by dividing $pv_-/\delta^2$ by $p = 1 - \omega_1^2 T_1[\mathbf{P}_0^{-1}]_{11}$ (Eq. (75)).

The columns in Fig. 3 reveal the effect of the g-tensor anisotropy on the different factors relevant to SE. $v_+/\delta^2$ in the third row of the figure is composed of two SE lines centered at $-\omega_I$ and $+\omega_I$. At the fastest tumbling (leftmost column), each of these two lines is symmetric and approximately Lorentzian. When the tumbling slows down, each line broadens and becomes asymmetric. At the slowest tumbling rate (rightmost column), each line resembles a powder EPR spectrum with anisotropic g

tensor. We see that in the regime of slow tumbling the profile of $v_+/\delta^2$ is no longer symmetric (i.e., even) with respect to the electronic resonance at zero offset frequency.

In the second row of Fig. 3 we show $v_-/\delta^2$ (orange line), which is also composed of two SE lines centered at $-\omega_I$ and $+\omega_I$, with the former flipped with respect to the horizontal axis. For comparison, in the second row we also plotted $v_+/\delta^2$ and $-v_+/\delta^2$ (dashed red lines). We see that, for all tumbling rates, the two SE lines comprising $v_-/\delta^2$ exactly match their counterparts in $v_+/\delta^2$.

The first row of Fig. 3 shows the electronic saturation under g-tensor anisotropy (we actually plot the "non-saturation" $p = 1 - s$). Because of the large $B_1$ used in the calculations ($B_1 = 5.5\,\mathrm{G}$) appreciable electronic saturation is achieved for all shown tumbling rates. From the perspective of the solid effect, it is noteworthy that the saturation is more localized to on-resonance conditions when the g-tensor anisotropy is averaged out by the tumbling, and spreads to larger off-resonance frequencies when the tumbling slows down. This spread broadens the saturation profile and reduces its maximum. However, in spite of the substantial increase of the spectral width of the saturation when going from $\tau_{\mathrm{rot}} = 2\,\mathrm{ns}$ to $\tau_{\mathrm{rot}} = 20\,\mathrm{ns}$, the maximum decreases only moderately, remaining close to 50% at the slower tumbling rate.

Of course, the amplitude of the saturation profile depends not only on $B_1$ but also on the electronic $T_1$ relaxation time. To illustrate this dependence, we recalculated all curves in Fig. 3 after increasing $T_1$ five-fold to $500\,\mathrm{ns}$. The result, which is shown in Fig. A1, demonstrates larger saturation for all tumbling rates. At the same time, $v_-/\delta^2$ and $v_+/\delta^2$ (second and third rows) remain entirely unaffected. This demonstrates that the SE lines do not experience the power-broadening that affects the EPR spectrum.

Finally, the last row of Fig. 3 shows $pv_-/\delta^2$ (solid green line), which equals the product of the first and second rows. From Eq. (7) we know that $pv_-/\delta^2$ basically gives the SE-DNP spectrum, up to an overall scaling factor. Since $pv_-$ is suppressed by the electronic saturation compared to $v_-$, we see that $pv_-/\delta^2$ is somewhat reduced at offsets between the canonical SE positions $\pm\omega_I$. Because both the electronic saturation profile and the profile of $v_-$ are asymmetric in the slow motional regime where the EPR line exhibits clear g-broadening, the line shape of the SE-DNP spectrum (proportional to $pv_-$) is no longer antisymmetric (i.e., odd) with respect to the electronic resonance. This is most visible for the green line in the lower rightmost corner of Fig. 3.

## 4.5  Solid effect in liquids

In the light of Sect. 3.3, the generalization to liquids consists of calculating the matrix

$$\mathcal{Q} = j_{11}(\mathcal{B}), \tag{93}$$

and using it instead of $\mathcal{B}^{-1}$ in Eq. (87):

$$
\begin{aligned}
R_{1I}^A &= \langle\delta^2\rangle \mathrm{Re}\{[[\mathcal{Q}_{\omega_1=0}]_{zz}]_{11}\} \\
v_+ &= \langle\delta^2\rangle \mathrm{Re}\{[[\mathcal{Q}]_{zz}]_{11}\} - R_{1I}^A \\
pv_- &= -\langle\delta^2\rangle \omega_1 \mathrm{Im}\{[([\mathcal{Q}]_{zx} + [\mathcal{Q}]_{zy}\mathbf{R}_2^{-1}\boldsymbol{\Delta})\mathbf{P}_0^{-1}]_{11}\}.
\end{aligned}
\tag{94}
$$

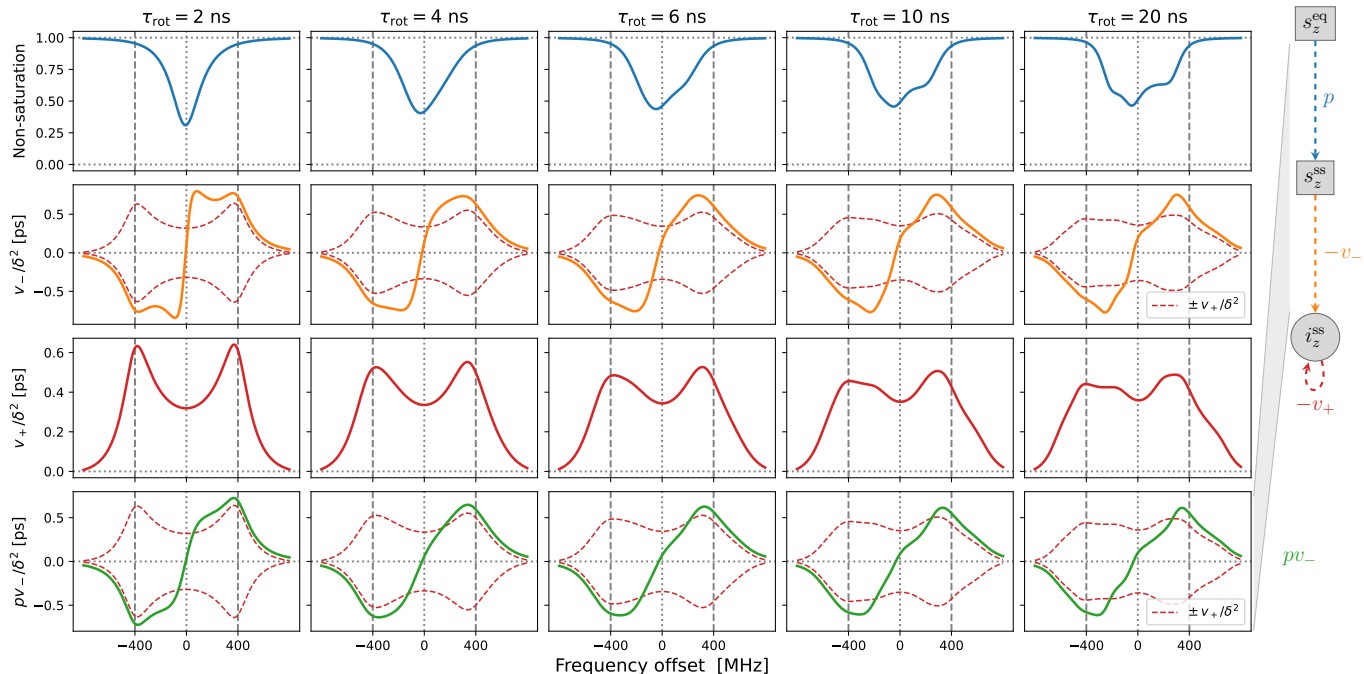

**Figure 4.** Same as Fig. 3 for the FFHS model of translational diffusion with $\tau_{\text{ffhs}} = 6$ ns.

Because the $zz$ sub-block of $\mathcal{B}$ is diagonal and does not couple to the rest when $\omega_1 = 0$, we deduce that

$$R_{1I}^A = \langle \delta^2 \rangle \text{Re}\{j_{11}(R_1 + i\omega_I)\}, \tag{95}$$

which is identical to Eq. (43). Thus, as we already observed for 'solids', the expression for $R_{1I}^A$ is not affected by the anisotropy of the g tensor and the slow tumbling of the radical.

In Fig. 4 we show the same properties as in Fig. 3 but now in the presence of translational diffusion treated by the FFHS model with motional time scale $\tau_{\text{ffhs}} = 6$ ns. Several changes compared to 'solids' (Fig. 3) are worth pointing out.

In line with our previous understanding (Sezer, 2023b), the SE lines comprising $v_+/\langle \delta^2 \rangle$ are broadened by the translational motion that modulates the dipolar interaction (red lines in the third row of Fig. 4). This motional broadening reduces their maximum intensities compared to 'solids' (Fig. 3, third row). Previously, in the case of Lorentzian lines, the reduction of intensity in the transition from solids to liquids was dramatic, by more than a factor of ten (Sezer, 2023b, Figs. 3, 4 and 5). In contrast, the reduction in the presence of g-tensor broadening is about a factor of two (compare third rows of Figs. 3 and 4). This observation may help rationalize why the maximum SE-DNP enhancement in liquids, e.g., about 50 for trityl in glycerol at 320 K (Kuzhelev et al., 2023), is not negligibly smaller compared to the enhancements that are obtained in the solid state. We also point out that, while reducing the maximum SE intensities in the vicinity of $\pm\omega_I$, the motional broadening substantially increases the intensities at the smaller offsets around the electronic resonance.

Besides the motional broadening, the progression from left to right in the third row of Fig. 4 demonstrates additional g-tensor broadening, which was present also in 'solids'. However, now the two SE lines are affected differently by the g-tensor anisotropy, making the profile of $v_+/\langle\delta^2\rangle$ at slow tumbling rates rather irregular.

Moving on to the second row in Fig. 4, we see that the SE lines that make up $v_-/\langle\delta^2\rangle$ (orange) are now completely different from their counterparts in $v_+/\langle\delta^2\rangle$ (dashed red). The increased intensity in the vicinity of the electronic resonance due to

590 motional broadening is also manifested by $v_-/\langle\delta^2\rangle$. For the fastest tumbling in the figure (leftmost column), the fluctuations of the dipolar interaction not only broaden the SE lines but also enable a new phenomenon, which is manifested as near-resonance peaks that are comparable in magnitude to the peaks at $\pm\omega_I$ but clearly distinct from them (orange line). These peaks reflect the multiplicative contribution of the dispersive EPR signal to $v_-$ (Sezer, 2023a, b). For faster translational diffusion the near-resonance peaks may become larger than the peaks at $\pm\omega_I$, as can be seen in the leftmost column of Fig. A3a (orange line).

Because they are more strongly suppressed by the electronic saturation, however, these peaks do not exceed the SE peaks in the final enhancement profile (Fig. A3a, leftmost column, green line).

Up to an overall scaling factor, the green lines in the last row of Fig. 4 correspond to the SE-DNP enhancement profile. Because its middle part is suppressed by the electronic saturation, this profile in the presence of g-tensor broadening becomes very non-symmetric and responds sensitively to the tumbling of the polarizing agent. To further illustrate the influence of

600 the electronic saturation on the SE-DNP spectrum, in Fig. A2 we show the same curves but calculated with five-fold longer electronic spin-lattice relaxation time ($T_1 = 500\,\text{ns}$), which leads to larger saturation. Similarly, to illustrate the effect of translational diffusion, we recalculated the curves in Fig. 4 for $\tau_{\text{ffhs}} = 3\,\text{ns}$ (two times faster) and $\tau_{\text{ffhs}} = 12\,\text{ns}$ (two times slower). The results are presented in Fig. A3. These additional simulations show that the SE-DNP line shape is very sensitive to the times scales of molecular motion.

In the next section, we systematically vary the degrees of power broadening and motional broadening to match the experimental DNP profiles from Fig. 1.

## 5 Disentangling the solid and Overhauser DNP effects

Using the developed methodology, we now analyze the experiments from Fig. 1. In the light of Eqs. (2) and (7) for the OE and SE enhancements, we will identify the profile of the electronic saturation (Fig. 4, first row) with $\epsilon_{\text{OE}}$ and the profile of

610 $pv_-/\langle\delta^2\rangle$ (Fig. 4, last row, green line) with $\epsilon_{\text{SE}}$. The tumbling times to be used in the DNP calculations will be obtained by fitting the experimental cw-EPR spectra. We start with 10-Doxyl-PC (Fig. 1a,c) as its experimental spectra were more amenable to unrestricted fits of all parameters.

### 5.1 Analysis of 10-Doxyl-PC

#### 5.1.1 Fit to the cw-EPR spectrum

Derivative EPR spectra were calculated from the first (i.e., 00th) components of the expressions in Eq. (77) for different values of the fitting parameters. In the fit, we varied the time scale of tumbling, $\tau_{\rm rot}$, as well as the g-tensor anisotropies $\gamma_0^2$ and $\gamma_2^2$ (Eq. (50)). As we have no precise knowledge of the field $B_0$ at the sample, we freely shifted the calculated spectra along the horizontal axis to achieve best match with experiment. Since this leaves one of the g-tensor components undetermined we took $g_{zz} = 2.0023$, which is typical for nitroxides.

The numerical integrals of the derivative EPR spectra in Figs. 1a and 1b (dotted-dashed blue lines) do not come down exactly to zero at the end of the integration range at high frequency offsets. This points to the possibility that the in-phase component, $s_y$, is mixed slightly with the out-of-phase component, $s_x$. To account for this possibility, we fitted the derivative EPR spectra by calculating

$$\frac{\partial s_y^{00}}{\partial \Delta}\cos\phi + \frac{\partial s_x^{00}}{\partial \Delta}\sin\phi, \tag{96}$$

where the angle $\phi$ controlled the degree of mixing.

All in all, not counting the shift along the horizontal axis, we had four fitting parameters: $\gamma_0^2$, $\gamma_2^2$, $\tau_{\rm rot}$ and $\phi$. The best fit to the cw-EPR spectrum of 10-Doxyl-PC is shown in Fig. 5a. The corresponding fitting parameters are given in the upper half of Table 1.

Encouragingly, our fitted spectrum shows rather good agreement with experiment, in spite of the simplifying assumptions of 630 the theoretical model, namely isotropic rotational diffusion and absence of hyperfine interaction. To check the effect of the latter on the cw-EPR spectrum, we used Easyspin (Stoll and Schweiger, 2006) to simulate spectra with our fitted parameters but now also including a nitroxide hyperfine tensor, $A = {\rm diag}(14, 14, 90)$ MHz. The result is given in Fig. A4a. The modification due to the hyperfine interaction, although small as expected at high magnetic fields, is clearly visible. Nevertheless, the comparison of the integrals of the cw-EPR spectra in Fig. A4b suggests that the error made by neglecting the hyperfine interaction when 635 calculating the DNP spectrum should be small.

Regarding the values of the fitted parameters, it was encouraging to see that the fit resulted in a negligibly small mixing angle of $\phi = -1.3°$, indicating that the measured spectrum correctly reflects the in-phase EPR component. With $B_0 = 9.4029$ T and $g_{zz} = 2.0023$, the fitted g-tensor anisotropies that are given in Table 1 implied

$$g_{xx} = 2.00755, \qquad g_{zz} = 2.00555. \tag{97}$$

These values are rather reasonable for a nitroxide spin label. Finally, the fitted time scale of rotational diffusion was $\tau_{\rm rot} = 5.2$ ns. For comparison, the same time scale for the nitroxide free radical TEMPOL in water is about 20 ps (Sezer et al., 2009). However, unlike TEMPOL, our spin label is covalently attached to the lipid chain.

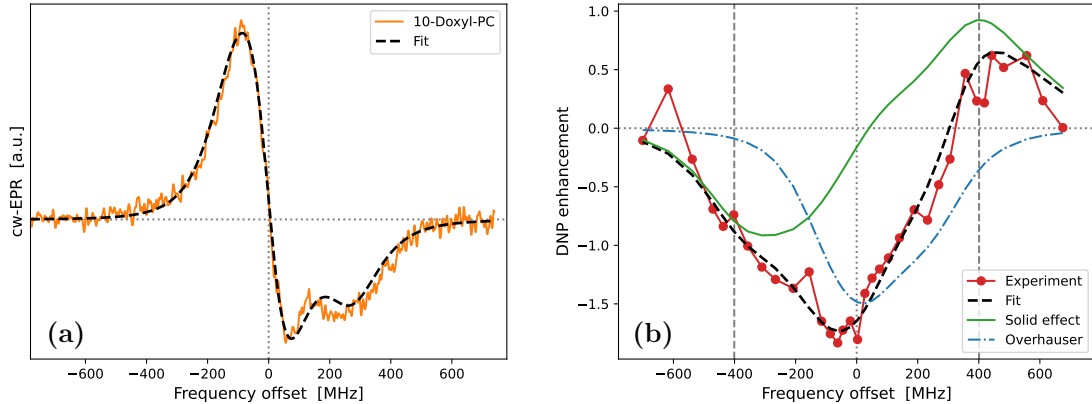

**Figure 5.** Fits to the experimental cw-EPR spectrum (**a**) and DNP spectrum (**b**) of 10-Doxyl-PC. In both cases, our best fits are shown with dashed black lines. The DNP spectrum in (**b**) is calculated by adding the contributions of SE (solid green line) and OE (dotted-dashed blue line), both of which are affected by the g-tensor anisotropy. The fit parameters are given in Table 1.

**Table 1.** Parameters obtained from the fits to the experimental data. $B_1 = 0.02\,\text{G}$ for EPR and $5.5\,\text{G}$ for DNP. Homogeneous $T_2^{\text{homog}} = 20\,\text{ns}$ was used for both EPR and DNP.

| fit | parameter | 10-Doxyl-PC | 16-Doxyl-PC | |
|---|---|---|---|---|
| EPR | $\gamma_0^2, \gamma_2^2$ (MHz) | $-373, 107$ | from 10-PC | |
| | $\tau_{\text{rot}}$ (ns) | 5.2 | 1.9 | |
| | $\phi$ (°) | $-1.3$ | $-2$ | |
| | shown in figure | 5a | 6a | |
| DNP | $\tau_{\text{ffhs}}$ (ns) | 6.4 | from 10-PC | 15.3 |
| | $T_1$ (ns) | 123 | 153 | 141 |
| | $\sigma_{\text{OE}}$ (-) | 2.43 | 2.385 | 2.57 |
| | $\sigma_{\text{SE}}$ (ps$^{-1}$) | 1.51 | 1.35 | 1.09 |
| | shown in figure | 5b | 6b | 7 |

### 5.1.2 Fit to the DNP spectrum

Fixing the g-tensor components and the tumbling time to the values obtained from the fit to the cw-EPR spectrum, we proceeded
to fit the DNP spectrum of 10-Doxyl-PC (Fig. 1c). In the calculations, we fixed the mw field to $B_1 = 5.5\,\text{G}$, which is our best estimate for the home-built Fabry-Pérot resonator operating at maximum power (Denysenkov et al., 2022). During the fits, we again allowed for global shift of the calculation along the horizontal axis. In addition, we fitted the electronic $T_1$ time, which has a direct effect on the electronic saturation profile, as well as the time scale of translational diffusion, $\tau_{\text{ffhs}}$, which is responsible for the motional broadening of the SE lines.

In the fit, we calculated the electronic saturation factor (Eq. (75)) and the time scale $pv_-(\Delta)/\langle\delta^2\rangle$ (last equality in Eq. (94)) as functions of the offset frequency $\Delta$. Up to unknown multiplicative factors, these correspond to, respectively, the OE and SE enhancement profiles (Eqs. (2) and (7)). We then fit the experimental DNP spectrum by calculating

$$\epsilon(\Delta) = \sigma_{\mathrm{OE}} \times s(\Delta) + \sigma_{\mathrm{SE}} \times \frac{pv_-}{\langle\delta^2\rangle}(\Delta), \tag{98}$$

where the scaling parameters $\sigma_{\mathrm{OE}}$ and $\sigma_{\mathrm{SE}}$ were also allowed to vary freely. As a result, not counting the shift along the
655 horizontal axis, our fit contained four fitting parameters: $\tau_{\mathrm{ffhs}}$, $T_1$, $\sigma_{\mathrm{OE}}$ and $\sigma_{\mathrm{SE}}$. The best fit to the DNP spectrum of 10-Doxyl-PC is shown in Fig. 5b. It is noteworthy how the total DNP enhancement (dashed black line) emerges from the sum of the SE (green line) and OE (dotted-dashed blue line) contributions. The corresponding fitting parameters are given in the bottom half of Table 1.

In the case of 10-Doxyl-PC, the intuitive analysis of Neudert et al. (2017) for identifying the OE and SE components of
660 a mixed DNP spectrum using the integrated cw-EPR line shape already performed very well (Fig. 1c). It is, therefore, not surprising that our analysis, which has more fitting parameters, agrees better with the experimental DNP spectrum (Fig. 5b). Both deficiencies of the intuitive approach, namely, too narrow OE and SE contributions due to the lack of, respectively, power broadening and motional broadening, appear to be satisfactorily addressed.

On a more fundamental level, our simulation shows that, due to the simultaneous power- and motional-broadening, the
665 OE and SE contributions to the DNP enhancement are not only rather asymmetric but also overlap extensively. It should, therefore, be practically impossible to extract any molecular information from the mixed DNP spectrum without a complex, quantitative analysis. In our specific case, the fit resulted in a translational time scale $\tau_{\mathrm{ffhs}} = 6.4\,\mathrm{ns}$, and suggested that the electronic relaxation time should be about $T_1 = 120\,\mathrm{ns}$. At the high magnetic field of the experiment ($B_0 = 9.4\,\mathrm{T}$) this spin-lattice relaxation time is practically impossible to measure in the liquid state.

In addition to $\tau_{\mathrm{ffhs}}$ and $T_1$, the fit to the DNP spectrum of 10-Doxyl-PC also produced the following numerical values for the two scaling parameters in Eq. (98): $\sigma_{\mathrm{OE}} = 2.4$ and $\sigma_{\mathrm{SE}} = 1.5\,\mathrm{ps}^{-1}$. These will be analyzed in Sect. 5.3 together with the corresponding values for 16-Doxyl-PC.

## 5.2 Analysis of 16-Doxyl-PC

Because the g-tensor anisotropies are largely averaged in the cw-EPR spectrum of 16-Doxyl-PC (Fig. 1b), we did not attempt
to fit them. Instead, we fixed all three components to the values obtained from 10-Doxyl-PC. This left only the rotational time, $\tau_{\mathrm{rot}}$, and the mixing angle, $\phi$, as fitting parameters, not counting the shift along the horizontal axis. As the automated fitting did not behave well, we varied these two parameters manually. One satisfactory fit, obtained with the parameters that are given in Table 1, is shown in Fig. 6a. We mention that the relative heights of the two lines in the calculation were slightly improved by using a small mixing angle of $\phi = -2°$.

Although, overall, the fit is not bad, the middle part of the calculated spectrum changes too sharply and its high-frequency line is too narrow compared to experiment. We again used Easyspin to check whether these deficiencies are due to the lack of hyperfine interaction. The spectra for $\tau_{\mathrm{rot}} = 1.9\,\mathrm{ns}$ with and without hyperfine interaction are shown in Fig. A5a. As the whole

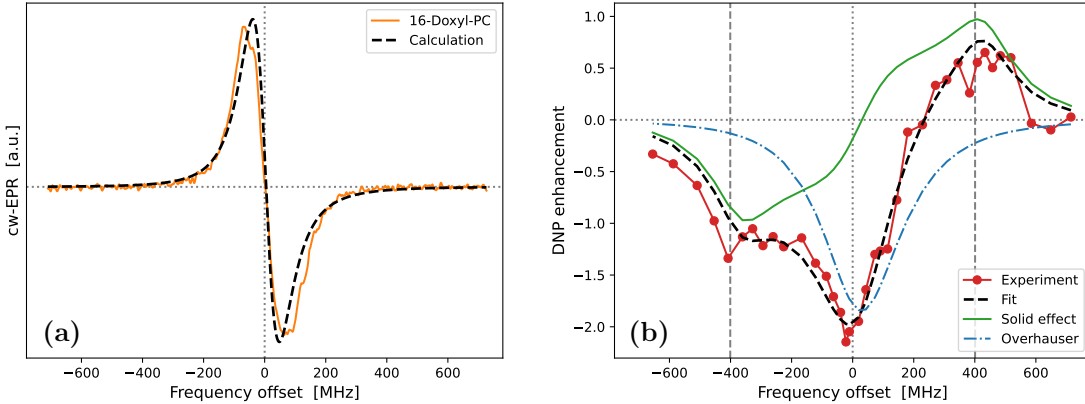

**Figure 6.** Same as Fig. 5 but for 16-Doxyl-PC. The fitted parameters are given in the second to last column of Table 1. Because all fitted lines in Figs. 5, 6 and 7 are calculated only at the experimental offsets, the green SE lines are not perfectly smooth.

spectrum is narrower than that of 10-Doxyl-PC, the effect of the hyperfine tensor is comparatively larger. Nonetheless, the integrated EPR lines in Fig. A5b show that the extra width due to the hyperfine tensor should not compromise our subsequent

analysis of the DNP spectrum, which will experience additional power-broadening and motional-broadening.

Moving on to the DNP spectrum, we observed that the free fit of all parameters resulted in $\tau_{\text{ffhs}}$ that was more than two times larger than that of 10-Doxyl-PC, as we explain below. Considering this to be unrealistic, we fixed $\tau_{\text{ffhs}}$ to the value that was obtained from 10-Doxyl-PC. Thus, not counting the horizontal translation of the calculated DNP spectrum, our automated fit had three fitting parameters: $T_1$, $\sigma_{\text{OE}}$ and $\sigma_{\text{SE}}$. The outcome is shown in Fig. 6b. The corresponding parameters are given in

the second-last column of the lower half of Table 1.

At 9.4 T the electronic Larmor precession time scale is about half a picosecond, which is three orders of magnitude less than the rotational time scales inferred from the cw-EPR spectra. On such sub-ps time scales, the local dynamics of the spin labels at positions 10 and 16 should not be very different from each other. Since the spin-lattice relaxation is determined by dynamics on the electronic Larmor time scale, we were satisfied that the fitted $T_1 = 150$ ns was close to that from 10-Doxyl-PC.

The performance of the simple analysis of Neudert et al. (2017) was poorer for 16-Doxyl-PC (Fig. 1d). Compared to it, our fit to the DNP enhancement profile is excellent (Fig. 6b). The only part of the DNP spectrum that our calculation systematically underestimates are the five leftmost experimental points. Although there are other individual experimental points that lie further from the calculated spectrum, these five points are persistently lower by about 0.2 enhancement units.

Observe that the downward shift of the fifth experimental point (together with the first four points) produces an enhancement

peak at around $-400$ MHz. The only way our automated fit can create a pronounced peak at this offset is by making the SE contribution (green line) more "solid-like", i.e., by increasing $\tau_{\text{ffhs}}$ and reducing the motional broadening. (The lower left corner of Fig. A3b provides an example of such more solid-like SE line shape.) We thus identify the systematic displacement of the leftmost five points to be responsible for the increase of $\tau_{\text{ffhs}}$ when it is allowed to vary freely during the fit.

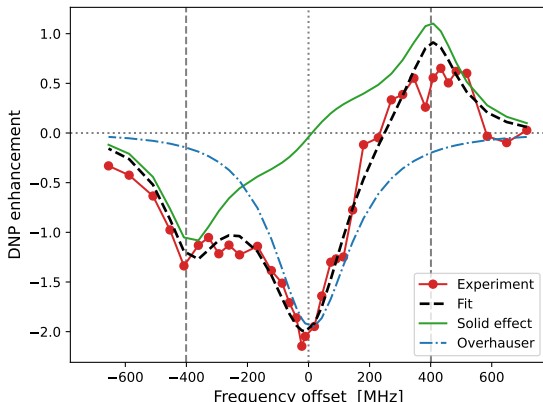

**Figure 7.** Same as Fig. 6b but also fitting $\tau_{\text{ffhs}}$. The fitted parameters are given in the last column of Table 1. Observe that the OE contribution to the DNP spectrum (dotted-dashed blue line) is narrower (i.e., more "liquid-like") than that of 10-Doxyl-PC (Fig. 5b), while the SE contribution is more "solid-like" because the time scale $\tau_{\text{ffhs}}$ is 2.4 times longer.

The best fit that we obtained when $\tau_{\text{ffhs}}$ was included among the other fitting parameters is shown in Fig. 7. (The resulting
fit parameters are given in the last column of the lower half of Table 1.) Indeed, with $\tau_{\text{ffhs}} = 15.3\,\text{ns}$, the SE lines (green) have become sharper and a small enhancement peak at $-400\,\text{MHz}$ has emerged (dashed black line). Although the enhancement around $+400\,\text{MHz}$ has been compromised in the process, the overall fit to all experimental points is improved compared to Fig. 6b.

The two alternative fits in Figs. 6b and 7 correspond to very different time scales of translational diffusion. Nevertheless,
within the variability of the measurements, they both agree with the DNP data. Considering the experimental challenges of liquid-state DNP at such high magnetic fields and large mw powers, further decreasing the experimental variability will be very hard. It is, therefore, important to analyze together several different experimental constructs, like our 10- and 16-Doxyl-PC. The final decision of which fit to the DNP spectrum of 16-Doxyl-PC is "better" can only be based on the overall consistency of the fitted parameters across all analyzed data. We return to this point in Sect. 5.3.

The other two parameters that emerged from the fit to the DNP spectrum of 16-Doxyl-PC were $\sigma_{\text{SE}}$ and $\sigma_{\text{OE}}$. These determine the amplitudes of the SE contribution (solid green lines in Figs. 6b and 7) and OE contribution (dotted-dashed blue lines) to the DNP enhancement (dashed black lines). We now turn to the analysis of these scaling parameters.

## 5.3 Additional molecular parameters

Ultimately, the motivation to disentangle a mixed DNP spectrum into its OE and SE components lies in the desire to extract
information about the molecular and spin properties that the respective DNP mechanism depends on. The main advantage of our procedure over the intuitive approach of Neudert et al. (2017) is that our decomposition produces physically interpretable parameters, like $\tau_{\text{ffhs}}$ and $T_1$. In addition, our scaling parameters $\sigma_{\text{OE}}$ and $\sigma_{\text{SE}}$ multiply, respectively, the saturation factor and $v_-/\langle \delta^2 \rangle$, whose absolute magnitudes are part of the calculation (Fig. 4, vertical axes). Thus, we can extract further information

**Table 2.** Analysis of the scaling parameters $\sigma_{\mathrm{OE}}$ and $\sigma_{\mathrm{SE}}$. Nuclear spin-lattice relaxation times with ($T_{1I}$) and without ($T_{1I}^0$) spin labels were measured at two different temperatures. These determine the leakage factor $f$. The coupling factor $c$ is obtained from $f$ and $\sigma_{\mathrm{OE}}$ using Eq. (99). The magnitude of the dipolar interaction responsible for SE ($\langle\delta^2\rangle$), obtained from $\sigma_{\mathrm{SE}}$ using Eq. (100), provides information about the effective contact distance ($b$). Combining $b$ with $\tau_{\mathrm{ffhs}}$ from Table 1, we estimate the diffusion constant of the FFHS model ($D_{\mathrm{ffhs}}$).

| Doxyl-PC | Temp. $T$ (K) | Nuclear $T_1$'s $T_{1I}^0$ (ms) | $T_{1I}$ (ms) | Overhauser $f$ | $\sigma_{\mathrm{OE}}$ | $c$ (‰) | Solid effect $\sigma_{\mathrm{SE}}$ (ps$^{-1}$) | $\langle\delta^2\rangle T_{1I}$ (ns$^{-1}$) | $b$ (nm) | $D_{\mathrm{ffhs}}$ (nm$^2$/μs) |
|---|---|---|---|---|---|---|---|---|---|---|
| 10 | 310 | 580 | 44 | 0.92 | 2.43 | 3.99 | 1.51 | 2.29 | 0.61 | 59 |
|    | 330 | 910 | 52 | 0.94 |      | 3.92 |      |      | 0.65 | 66 |
| 16 | 310 | 580 | 93 | 0.84 | 2.385 | 4.32 | 1.35 | 2.05 | 0.81 | 104 |
|    | 330 | 910 | 120 | 0.87 |      | 4.17 |      |      | 0.89 | 123 |
| 16* | 310 | " | " | " | 2.57 | 4.65 | 1.09 | 1.66 | 0.87 | 50 |
|     | 330 | " | " | " |      | 4.50 |      |      | 0.95 | 59 |

from the fitted values of $\sigma_{\mathrm{OE}}$ and $\sigma_{\mathrm{SE}}$. In contrast, because the simple approach rescales the integrated cw-EPR spectrum whose amplitude is arbitrary, the values of its scaling factors are not informative.

Using Eq. (2) for the OE enhancement, the coupling factor $c$ is readily expressed in terms of $\sigma_{\mathrm{OE}}$:

$$c = \frac{\sigma_{\mathrm{OE}}}{f}\frac{\gamma_I}{|\gamma_S|}, \tag{99}$$

where the leakage factor $f$ can be obtained by measuring the nuclear spin-lattice relaxation times (Eq. (4)).

We measured the $T_1$ values for the chain protons of DOPC (without spin-labeled lipids) at 310 K and 330 K using the Fabry-Pérot probe. These are given in the $T_{1I}^0$ column of Table 2. Additionally, we measured the nuclear spin-lattice relaxation times in the presence of either 10- or 16-Doxyl-PC (column $T_{1I}$ of Table 2). The target temperature of the DNP experiments (320 K) lies between the two temperatures at which the nuclear $T_1$ times were measured. However, considering the possibility of mild temperature rise by several degrees, we expect the values at 330 K to closely reflect the DNP conditions. Nonetheless, we carry out the following analysis using the $T_1$ values measured at both 310 K and 330 K.

The leakage factors obtained from Eq. (4) are shown in the column $f$ of Table 2. Using the values of $\sigma_{\mathrm{OE}}$ from Table 1 in Eq. (99), we arrived at the coupling factors in column $c$ of Table 2. In the case of 16-Doxyl-PC, the analysis was performed for the fit where $\tau_{\mathrm{ffhs}}$ was fixed at 6.4 ns (denoted 16 in Table 2) as well as for the fit where $\tau_{\mathrm{ffhs}}$ was free to change (denoted 16*). (These two alternatives correspond to the last two columns of Table 1.) For both choices, somewhat larger coupling factors were deduced for 16-Doxyl-PC compared to 10-Doxyl-PC. The estimated coupling factors are less than two times smaller than what we have obtained previously for TEMPOL in DMSO, and about four times smaller than the coupling factors between TEMPOL and the protons of toluene (Prisner et al., 2016; Sezer, 2013; Küçük et al., 2015).

Turning now to SE, using the enhancement in Eq. (7) we express the unknown strength of the dipolar interaction in terms of the scaling parameter $\sigma_{\text{SE}}$ as follows:

$$\langle \delta^2 \rangle T_{1I} = \sigma_{\text{SE}} \frac{\gamma_I}{|\gamma_S|}. \tag{100}$$

The values of $\langle \delta^2 \rangle T_{1I}$, which were calculated from the right-hand side of Eq. (100), are about $2 \, \text{ns}^{-1}$ for 10, 16 and 16* (Table 2). Since $v_+ / \langle \delta^2 \rangle$ is about 1 ps (Fig. 4, third row), we conclude that $v_+ T_{1I} \ll 1$, which justifies our use of the approximation in Eq. (7) throughout the analysis, including during the fit to the DNP spectra.

From the expression of $\langle \delta^2 \rangle$ (Eq. (39)), we can write the contact distance of the translational FFHS model as

$$b^3 = N \frac{2\pi}{5} D_{\text{dip}}^2 \frac{T_{1I}}{\sigma_{\text{SE}}} \frac{|\gamma_S|}{\gamma_I}, \tag{101}$$

where $N$ is the number density of the electronic spins. Since, in principle, all parameters on the right-hand side of Eq. (101) are measurable, we can determine $b$. To estimate $N$, we note that the molecular volume of DOPC is $1.3 \, \text{nm}^3$ (Greenwood et al., 2006). Since there are 20 unlabeled lipids for one labeled one, we estimate $N = (20 \times 1.3 \, \text{nm}^3)^{-1}$, which corresponds to a molar concentration of 64 mM. Using this number in Eq. (101), we obtained the values of $b$ that are given in the second to last column of Table 2.

When the values of $b$ are interpreted literally as "contact distance" between the nitroxide spin label and the protons of the lipid chains, their substantial variation between 10- and 16-Doxyl-PC is disturbing. From that perspective, it is clear that the parameter $b$ of the FFHS model, which we used to account for the fluctuations of the dipolar interaction due to molecular translations, cannot reflect the actual molecular distances of closest approach.

Because $b$ was obtained from the scaling parameter $\sigma_{\text{SE}}$, only information about the *amplitude* of the SE enhancement has been directly used in its estimate. In contrast, the motional time scale $\tau_{\text{ffhs}}$ (Table 1) encodes information about the *line shape* of the SE enhancement. From these complementary features of the SE contribution to the DNP spectrum, we have managed to determine both $b$ and $\tau_{\text{ffhs}}$. Having access to these two parameters, we can calculate the diffusion constant of the FFHS model from Eq. (38). The results are given in the last column of Table 2. To our surprise, we obtained very similar values for 10 and 16*, while the diffusion constant for 16 is two-fold larger. (Given the variability in the experimental data and the fact that the fits to the DNP spectra are not unique, the differences between $D_{\text{ffhs}}$ of 10 and 16* should not be seen as meaningful.)

In an effort to identify a potential candidate for the physical motion that the FFHS model emulates, we observed that the coefficients of lateral translational diffusion for DOPC in oriented bilayers are $20 \, \text{nm}^2 \, \mu\text{s}^{-1}$ at 323 K, and $26 \, \text{nm}^2 \, \mu\text{s}^{-1}$ at 333 K (Filippov et al., 2003, Fig. 6a). These, we expect, bracket the value at our DNP conditions. The diffusion in the FFHS model corresponds to the *relative* translation of the nuclear and electronic spins, i.e., $D_{\text{ffhs}} = D_I + D_S$. Assuming that the lateral diffusion of spin-labeled PSPC in a DOPC bilayer is similar to that of DOPC, from the measured values given above we would expect $D_{\text{ffhs}}$ between 40 and $52 \, \text{nm}^2 \, \mu\text{s}^{-1}$. This range is surprisingly close to the estimates of 10 and 16* in the last column of Table 2, which suggests that the FFHS model in our analysis likely accounts for the lateral diffusion of the lipids in the plane of the bilayer.

Since it leads to a diffusion constant that is similar to (i) the known lateral diffusion of DOPC and (ii) the estimate obtained for 10-Doxyl-PC, we conclude that the fit to the DNP spectrum of 16-Doxyl-PC that is shown in Fig. 7 (i.e., the one that led

to "unreasonably" large $\tau_{\text{ffhs}}$) is more realistic than the one with fixed $\tau_{\text{ffhs}}$ (Fig. 6b). From the perspective of the diffusion constant, the longer motional time scale of 16* compared to 10, which resulted in more solid-like SE line shape with less motional broadening, reflects the fact that the "contact distances" in the two cases are different. In retrospect, it is amazing how the independent estimates of $b$ and $\tau_{\text{ffhs}}$ combine to yield practically identical diffusion constants for the two spin-labeling positions.

At the moment, it is not clear to us how to properly interpret the different values of $b$ at positions 10 and 16. Atomistic molecular dynamics simulations (Oruç et al., 2016) could, in principle, be used to investigate whether these effective contact distances reflect differences in proton density along the normal of the lipid bilayer, or arise for some other reason.

## 5.4  Limitations of the modeling

The calculated DNP spectra of 10-Doxyl-PC (Fig. 5b) and 16-Doxyl-PC (Fig. 7) agree well with the experiments, in spite of our simplistic treatment of the quantum and classical dynamics. Specifically, when modeling the spin dynamics, (i) we completely neglected the hyperfine interaction with the nuclear spin of $^{14}$N, which is present in nitroxide spin labels. In the case of the classical dynamics, (ii) we modeled the reorientation of the spin labels at positions 10 and 16 of the lipid chain as free, isotropic diffusion, and (iii) we modeled the dynamics of the acyl protons relative to the unpaired electron as isotropic translational diffusion that extends to infinity in all three spatial directions. We now comment on these deficiencies of the modeling.

Starting with the third point, it is clear that the translational diffusion of the polarized aliphatic protons (as well as that of the chain-attached spin labels) must be confined to the interior of the lipid bilayer and should not extend arbitrarily far along the direction perpendicular to the bilayer plane. In contrast, the FFHS model whose analytical correlation function we used in the calculations assumes isotropic diffusion in all spatial directions. To properly address this deficiency of the modeling, one would need to solve the diffusion equation with boundary conditions that reflect the confining planar geometry of the lipid bilayer, and then calculate the dipolar correlation function for such confined diffusion (preferably in closed, analytical form). In the meantime, one could argue that, because the dipolar interaction drops rapidly with distance, an overwhelming contribution to the dipolar correlation function should come from configurations in which the electron and nucleus are close to each other. In that case, the unphysical configurations that place the acyl protons outside the plane of the lipid bilayer (but are allowed in the FFHS model) may contribute relatively little. To support this argument, we observe that the numerical values of the FFHS parameter $b$ in Table 2 indicate that the shortest relevant distances for SE are about 0.6 nm (for 10-Doxyl-PC) and 0.9 nm (for 16-Doxyl-PC). These are three to five times smaller than the hydrophobic thickness of the DOPC lipid bilayer, which is about 3 nm (Kučerka et al., 2008).

Regarding the second deficiency, the problem here is that the nitroxide spin label is covalently fused to the lipid chain, thus its possible orientations should reflect the preferred alignment of the chain in the hydrophobic core of the bilayer. Furthermore, the fused nitroxide is not expected to have identical diffusion rates for rotations about different spatial directions. Clearly, both of these aspects (i.e., the orientational preference and the anisotropy) are missing from the free, isotropic rotational diffusion that we implemented. It is, however, known how to account for them in a rigorous and efficient way. Indeed, the MOMD

(microscopic order macroscopic disorder) model from the Freed lab (Meirovitch et al., 1984) treats anisotropic rotational diffusion in a restoring potential. In fact, this model has been extensively used to simulate high-field cw-EPR spectra of lipid bilayers by Freed (Lou et al., 2001; Costa-Filho et al., 2003) and Marsh (Livshits et al., 2004, 2006). The studies of Marsh and colleagues have focused on DMPC lipid bilayers containing appreciable amount of cholesterol, which puts them in a liquid-ordered phase. For DMPC with 40 mol % cholesterol at 30°C, 10-Doxyl-PC was deduced to be aligned with the director

(i.e., the direction normal to the bilayer plane) with an order parameter $S = 0.67$ (Livshits et al., 2004). If the orientational motion is imagined as being confined to a cone (Lipari and Szabo, 1982), this order parameter would correspond to a maximum possible deviation from the director of $\theta_0 = 40°$ in all directions. The lipid bilayers in our experiments are composed of pure DOPC lipids and are in their liquid-crystalline phase, where the ordering is substantially reduced. The liquid-crystalline phase of pure DPPC lipid bilayers has been characterized in the studies of Freed and colleagues. The order parameter reported for

16-Doxyl-PC in pure DPPC at 50°C is $S = 0.16$ (Costa-Filho et al., 2003). It corresponds to a maximum possible deviation from the director of $\theta_0 = 75°$, assuming the diffusion is confined to a cone. Although 10-Doxyl-PC is expected to be more ordered than 16-Doxyl-PC, it is not clear how much smaller than $\theta_0 = 75°$ its corresponding cone angle would be. (Because $S$ is the expectation value of a rank-2 spherical harmonic, the free rotational diffusion that we use corresponds to $\theta_0 = 90°$.) From these studies we conclude that the MOMD model (with an axial diffusion tensor) will likely improve our fits to the

experimental cw-EPR spectra. Nevertheless, free rotation may still be a good first approximation to the orientational dynamics of 10- and 16-Doxyl-PC in the liquid-crystalline phase of our lipid bilayers.

We should emphasize that our aim in the current paper is to show how to account for the rotational dynamics of the polarizing agent in the calculation of SE-DNP. In this context, we observe that while the cw-EPR spectra in derivative mode (Figs. A4a and A5a) are extremely sensitive to the details of the rotational motion of the radical, their integrals (Figs. A4b and A5b) are

830 much more forgiving. When contributing to the DNP spectrum these integrated EPR line shapes are additionally broadened by mw power (OE) and translational diffusion (SE) (Figs. 5b and 7). All these factors are expected to reduce the sensitivity of the DNP spectrum to the details of the radical tumbling (at least in comparison to the sensitivity of the cw-EPR line shape). We therefore think that, for the purposes of fitting the DNP spectrum, further improving the fit to the cw-EPR spectra at the cost of introducing more fitting parameters is not really justified. That being said, we stress that the formalism of Sect. 4 can be

straightforwardly extended to anisotropic diffusion in an orienting potential (i.e., the MOMD model). This would lead to larger matrices $\mathbf{R}_1$, $\mathbf{R}_2$ and $\mathbf{\Delta}$, whose matrix elements would be different than the expressions we gave in Sect. 4.2 for free, isotropic rotational diffusion.[5] Once correctly formed, these three matrices can be directly used in Eqs. (92) and (94) to calculate the SE-DNP spectra in, respectively, 'solids' and liquids.

Moving on to the first deficiency mentioned above, we remind the reader that we describe the SE spin dynamics in terms

of two sets of Bloch equations that are connected in series (Sezer, 2023a). These are the classical Bloch equations with

---

[5]The orienting potential will mix coefficients with different values of $L$, which will result in non-diagonal $\mathbf{R}_1$ and $\mathbf{R}_2$. The anisotropic rotation will mix coefficients whose $M$ indices differ by $\pm 1$, so odd values of $M$ will also need to be included. Finally, since the potential is defined with respect to the director axis, which may differ from the axes of both the laboratory frame and the molecular frame, it will be necessary to consider non-zero values of the index $N$ of the coefficients $\bar{s}_{MN}^L$. Clearly, for a given $L_{\max}$, the resulting matrices $\mathbf{R}_1$, $\mathbf{R}_2$ and $\mathbf{\Delta}$ will be substantially larger. A detailed presentation can be found in Schneider and Freed (1989) and Polimeno and Freed (1995).

Bloch matrix $\mathbb{B}_0$ (Sect. 3.1) and the "new Bloch equations" (Eq. (21)) with matrix $\mathbb{B} = \mathbb{B}_0 + i\omega_I$ (Sect. 3.2). Because our description of SE-DNP is based on Bloch equations, in Secs. 4.1 and 4.2 we reformulated Freed's treatment of slow tumbling as a generalization of the classical Bloch equations, such that the scalar elements of $\mathbb{B}_0$ became matrices in the space of the angular-momentum indices $LM$. The result was the "expanded Bloch matrix" $\mathcal{B}_0$ in Eq. (63). For this reformulation to work,
however, we had to neglect the hyperfine interaction, which is in fact treated by Freed et al. (1971). As a result, our analysis is formally deficient for nitroxide radicals. Nevertheless, we reasoned that it should be possible to illustrate the theoretical formalism in its current form by focusing on nitroxides at high magnetic fields, where the hyperfine interaction is expected to be negligible compared to the anisotropy of the g tensor. From this perspective, it should be clear that the DNP experiments that we analyzed here had been carefully selected.

Figures A4 and A5 show our attempt to assess the contribution of the neglected hyperfine interaction to the (integrated) EPR spectra of 10- and 16-Doxyl-PC. A somewhat more detailed analysis is contained in our response to the reviewers, which is freely accessible online. There we observe that the hyperfine interaction slightly broadens the EPR line of 16-Doxyl-PC (which is also visible in Fig. A5a). Since the only mechanism of broadening in our case is the rotational tumbling, our choice of $\tau_{\mathrm{rot}} = 1.9$ ns (Table 1) likely compensates for some of the "missing" hyperfine broadening. Such compensation does not
appear to be happening in the case of 10-Doxyl-PC, where the hyperfine interaction changes the shape but not the width of the EPR line (Fig. A4a). Ultimately, for the theory to be applicable to SE-DNP with nitroxide polarizing agents at lower magnetic fields, like X band (Gizatullin et al., 2021a, b), the description of the spin dynamics will need to include the nuclear spin of $^{14}$N. Since the dimension of the resulting Liouville space would need to increase by a factor of nine, it should be possible to preserve the two sets of connected Bloch equations after replacing each of their scalar matrix elements by a $9 \times 9$ matrix. Alternatively,
the two sets of Bloch equations should be replaced by the corresponding equations of motion for the density matrix in Liouville space. However, considering the inherent experimental uncertainty of the DNP enhancements that we compare with (Figs. 1c and 1d) and the achieved agreement between simulation and experiment (Figs. 5b and 7), we believe that such more complex modeling is presently not justified.

## 6 Conclusion

Once the spin dynamics of the solid effect has been formulated in time domain (Sezer, 2023a), it becomes possible to interface this quantum dynamics with various types of classical dynamics. The classical dynamics in Sezer (2023b) was the translational diffusion of the spins in a liquid; here we additionally included the rotational diffusion of the polarizing agent. To illustrate the practical utility of the resulting formalism, we analyzed either previously published (Sezer, 2023b) or previously unpublished (current paper) experimental DNP data on lipid bilayers. In our analysis, the treatment of molecular translation and rotation
was limited to the simplest possible models of free, isotropic diffusion. Surprisingly, in spite of the spatial anisotropy that one expects for hydrated lipid bilayers, previously we found that isotropic translation, as described by the FFHS model, worked well for the free radical BDPA in DMPC bilayers (Sezer, 2023b; Kuzhelev et al., 2022). Similarly, in the current paper we

found that the simplest treatment of free, isotropic rotation (together with FFHS translation) reproduced well the DNP field profiles of nitroxide-labeled lipids in DOPC bilayers.

DNP experiments with nitroxide free radicals in viscous liquids invariably manifest a mixture of SE and OE (Leblond et al., 1971b; Neudert et al., 2017). As these two DNP mechanisms are sensitive to molecular motions on vastly different time scales, it should be possible to obtain rich dynamical information by analyzing their contributions to the overall DNP enhancement. Disentangling the SE and OE contributions, however, has proven to be challenging (Leblond et al., 1971a). Here we fitted liquid-state DNP spectra by calculating enhancements that were affected by both the translational diffusion of the spins and the

rotational diffusion of the free radical. Since different motions modify the amplitude and the shape of the DNP spectrum in a highly concerted manner, by fitting the entire line shape of the enhancement we also gained access to the absolute magnitudes of the SE and OE contributions.

Our current treatment of SE-DNP in liquids uses only the correlation function of the dipolar interaction to describe the translational motion of the spins (Sezer, 2023b). This is formally correct only when the diffusion is much faster than the

885 nuclear $T_1$ relaxation. It should be possible to relax this condition and model slower spin diffusion, as relevant for SE in the solid state.

*Code and data availability.* The analyzed experimental data and the code used to generate the figures in the manuscript are available at https://github.com/dzsezer/solidDNPliquids_g-tensor (https://doi.org/10.5281/zenodo.8360325).

## Appendix A:  Additional figures

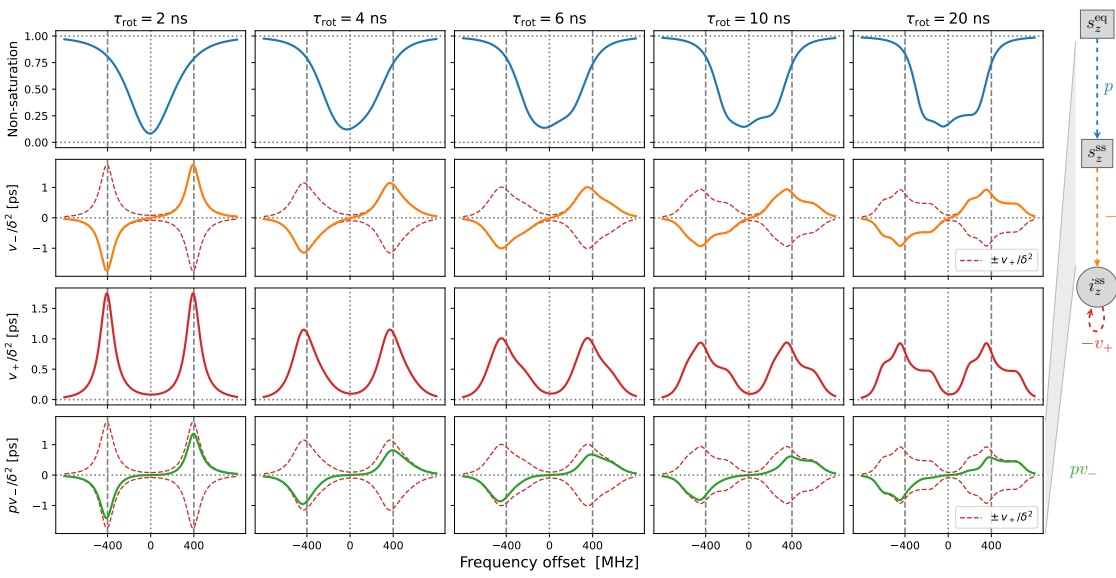

**Figure A1.** Same as Fig. 3 but with $T_1 = 500\,\text{ns}$ (i.e., five-fold longer), which leads to larger saturation of the allowed electronic transition. Only the first and last rows are affected.

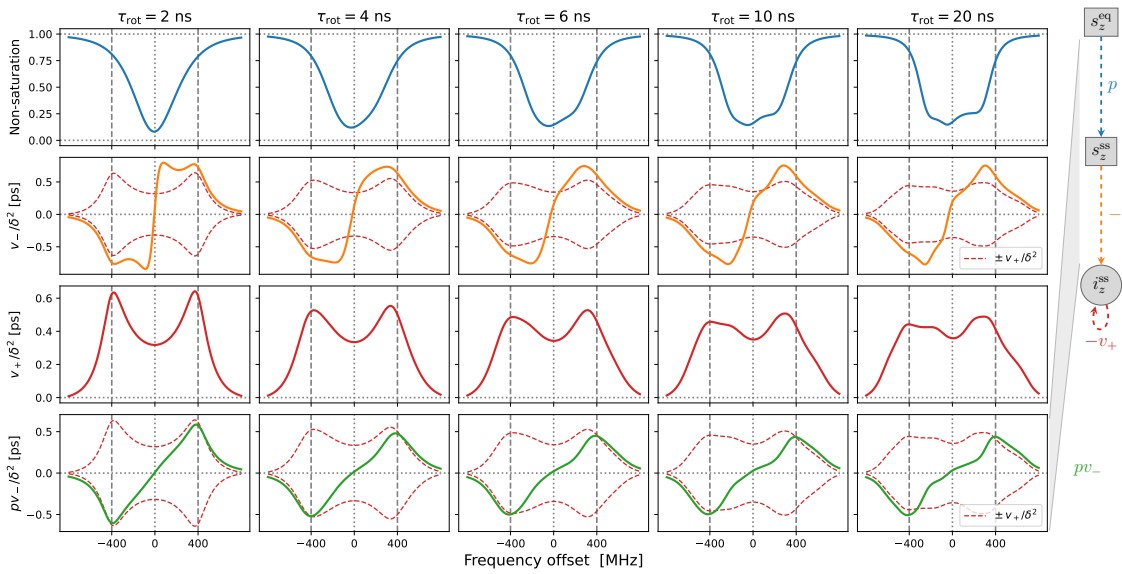

**Figure A2.** Same as Fig. 4 but with $T_1 = 500\,\text{ns}$, which leads to larger electronic saturation. As in the case of 'solids', only the first and last rows are affected.

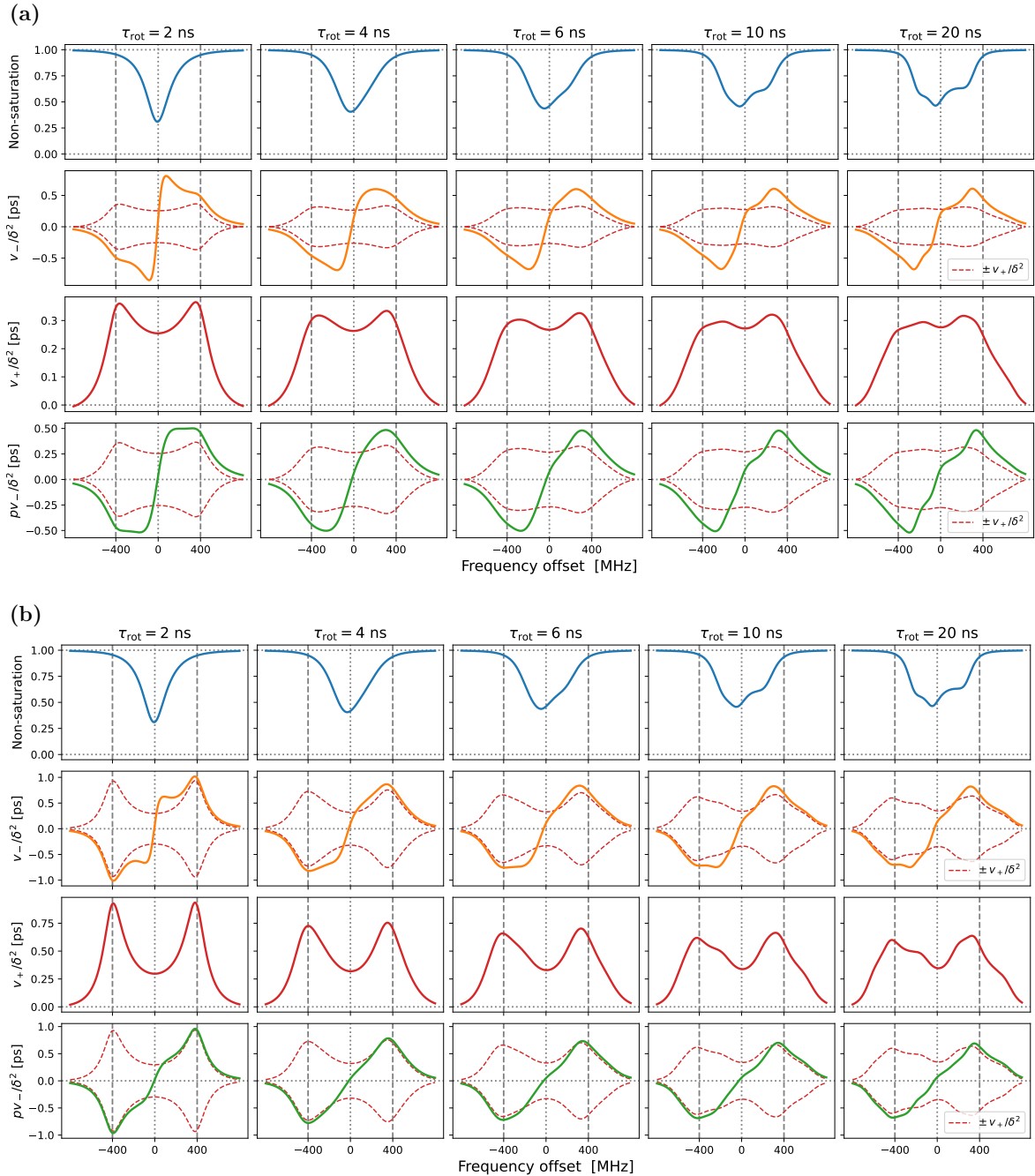

**Figure A3.** Same as Fig. 4 but with (**a**) $\tau_{\mathrm{ffhs}} = 3\,\mathrm{ns}$, i.e., two-fold faster translational motion which broadens the SE lines to a larger extent, and (**b**) $\tau_{\mathrm{ffhs}} = 12\,\mathrm{ns}$, i.e., more solid-like behavior. Observe how the predicted SE-DNP line shape (green line in the last row) responds sensitively to the time scale of the translational motion that is responsible for averaging the dipolar interaction.

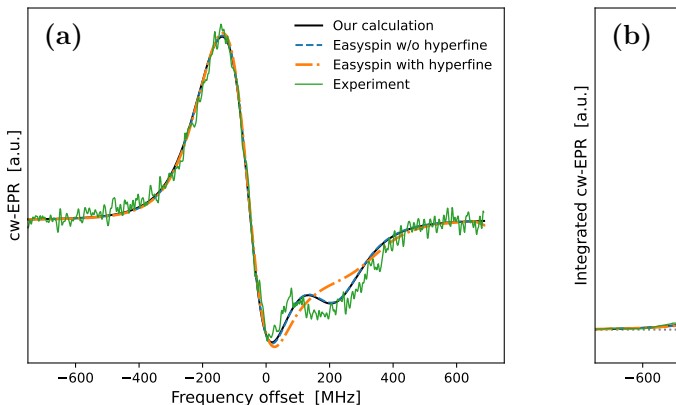
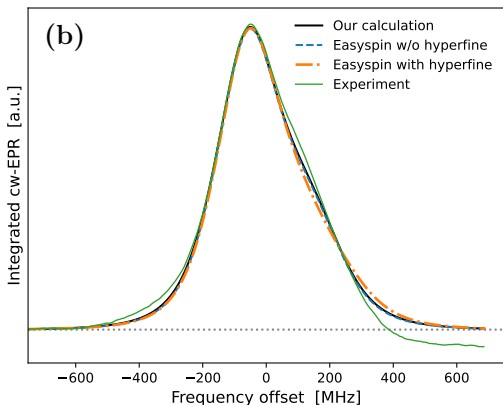

**Figure A4.** Effect of hyperfine tensor on the calculated EPR spectrum of 10-Doxyl-PC. Various derivative cw-EPR spectra (**a**) and their numerical integrals (**b**) are compared with each other. Our calculation (solid black line) agrees perfectly with the Easyspin (Stoll and Schweiger, 2006) simulation without a hyperfine tensor (dashed blue line). Including a hyperfine tensor with components $(14, 14, 90)$ MHz in the Easyspin calculation (dotted-dashed orange line) leads to visible changes in the derivative cw-EPR spectrum. However, the difference of the integrated EPR lines with and without a hyperfine tensor in (**b**) should be negligible as far as the simulation of the DNP spectrum is concerned.

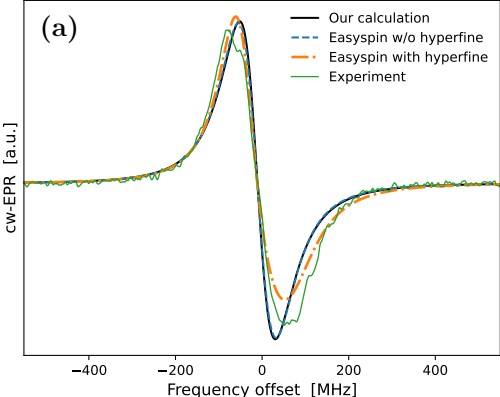
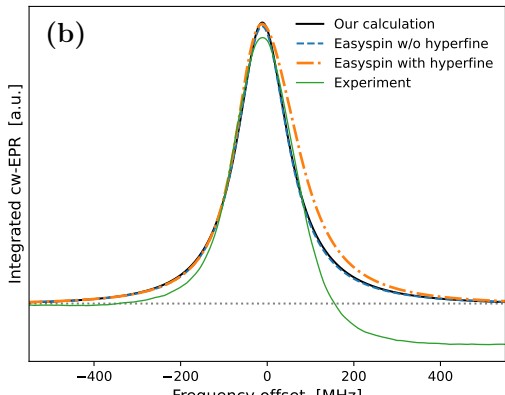

**Figure A5.** Same as Fig. A4 but for 16-Doxyl-PC. Because the cw-EPR spectrum is narrower to begin with, the relative contribution of the hyperfine tensor with components $(14, 14, 90)$ MHz is larger than in the case of 10-Doxyl-PC. Considering that the EPR line will experience additional power-broadening and motional-broadening in DNP, it should still be possible to safely neglect the extra width that the hyperfine tensor brings to the integrated EPR line in (**b**).

*Author contributions.* TFP envisioned the presented high-field DNP experiments in lipid bilayers and acquired funding for their execution. DD carried out these EPR and NMR experiments, deconvoluted the measured NMR signals to calculate the presented DNP spectra for the lipid chain protons, and performed the reported Easyspin simulations. DS conceived of the presented analysis of the experimental data, developed the reported theoretical and numerical framework, analyzed the data, and wrote the manuscript with feedback from all coauthors.

*Competing interests.* TFP is an associate editor of Magnetic Resonance.

*Acknowledgements.* We are grateful to Vasyl Denysenkov, without whom the high-field DNP experiments analyzed here would have not been possible. Andrey Kuzhelev was the first to realize that the DNP spectra presented in this paper manifest not only the Overhauser effect but also the solid effect.

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
