# Peer review of "The solid effect of dynamic nuclear polarization in liquids II: Accounting for g-tensor anisotropy at high magnetic fields"

_Magnetic Resonance, 2023_

## Author Comment (AC1)

Response to the referee comments of the manuscript
**The solid effect of dynamic nuclear polarization in liquids II: Accounting for g-tensor anisotropy at high magnetic fields**

Deniz Sezer, Danhua Dai, Thomas F. Prisner

September 14, 2023

We thank the reviewers for their insightful comments and suggestions which helped us improve the manuscript. In the following, the referee comments are reproduced in black, our answers are in blue, and the new additions to the manuscript are highlighted in red.

**1  Reviewer 3: Anonymous**

The manuscript describes detailed work on simulating DNP features originating from the solid effect, both in solids, and in liquids. The liquids case is the most interesting one, whereby asymmetry in the g-tensor leads to sizable solid-effect features. The work is truly impressive, providing simplified equations that allow very efficient computation of lineshapes, and thereby also providing an efficient data fitting modality. The work is probably easily accessible to the expert, but the broader readership would appreciate if further helpful comments were included. For example, in each section where a certain calculation is performed, it would be very helpful if at the beginning it were stated what the final goal of this portion is. I also feel that it may be suitable to cite the review article by Atsarkin. Some symbols are introduced in an ad hoc fashion which makes it harder to follow the thought process. For example, the provenance of the $i\omega_I$ terms in Eq. 19 is not explained, and one is left to guess. So some additional guiding statements would greatly improve the article.

- We have now cited Atsarkin's 2011 review that appeared in the Journal of Physics.

- We agree that Sec. 3, which summarizes the analytical results of our previous papers, was poorly motivated. To better prepare the reader for the calculations that follow, we have now introduced the following explanatory paragraphs at the beginning of this section.

  The classical Bloch equations describe the dynamics of the electronic magnetization, including under saturating conditions. In Sec. 3.1 we recall the relationship between the steady-state solution of the Bloch equations and cw-EPR. Then, in Sec. 3.2, closed-form expressions are obtained for the rate constants of the forbidden transitions that are driven by the microwaves in SE-DNP. These expressions, derived in this form for the first time (Eqs. (33) and (34)), are similar to the steady-state solutions of the Bloch equations but additionally contain (i) the strength of

the electron-nucleus dipolar interaction and (ii) the Larmor frequency of the polarized nuclear spin [1]. Finally, in Sec. 3.3 we remind the reader how these expressions should be modified in the presence of random modulation of the dipolar interaction, as relevant for liquids [2].

The reviewed results, which apply to a single Lorentzian line, will be extended in Sec. 4 to the regime of slow radical tumbling and anisotropic g tensor. In the process, some of the scalar variables that appear below, like the offset frequency and the electronic relaxation rates, will be replaced by square matrices, as we explain in Secs. 4.1 and 4.2. The generalization of Secs. 3.1, 3.2 and 3.3 along these lines is carried out in, respectively, Secs. 4.3, 4.4 and 4.5.

- The following clarifying sentence is now inserted at the end of Sec. 3.1:

  The $i\omega_I$ that has been added to the main diagonal of the matrix $\mathbb{B}$ will be needed for the dynamical description of the solid effect (see Sec. 3.2).

**2   Reviewer 2: Anonymous**

The paper is a continuation of the series of theoretical works on the Solid Effect. The paper considers the effect of g-factor anisotropy on the DNP spectra. I believe that this theoretical work deserves to be published in the "Magnetic Resonance". The work is rather bulk, both in volume and in the cases and approximations considered, and, in my opinion, it would be better to divide it into two publications, for instance, in one to consider the theory and in the second to compare the theory with experiment. But this is at the discretion of the authors. I assume that specialists working in the field of DNP will still be able to understand this paper in detail.

Similar to our first paper on the solid effect in liquids, the main aim of the current paper is to demonstrate that, once the spin dynamics of the solid effect has been formulated in time domain, it becomes possible to interface this quantum dynamics with various types of classical dynamics. The classical dynamics that was considered previously (Paper I) [2] was the translational diffusion of the spins in a liquid. In this manuscript we additionally include the rotational diffusion of the polarizing agent.

While our main goal has been to draw attention to the hitherto unexplored potential of the time-domain description of the solid effect, in both papers we demonstrate the practical utility of the formalism through the analysis of either previously published (Paper I) or previously unpublished (current paper) experimental DNP data. The fact that both sets of analyzed DNP experiments happen to be on lipid bilayers at $9.4\,\mathrm{T}$ (but with different polarizing agents) stems from the recent focus of the Prisner lab on DNP in lipid bilayers (DFG grant 405972957).

Given our main aim, the theoretical treatment of molecular translation and rotation has been limited to the simplest possible models of free, isotropic diffusion. Surprisingly, in spite of the spatial anisotropy that one expects for a hydrated lipid bilayer, in the previous/current paper we found/find that isotropic translation, as described by the force-free hard-sphere (FFHS) model, worked/works well. Similarly, in the current paper we find that the simplest treatment of free, isotropic rotation (together with FFHS translation) reproduces well the studied DNP field profiles. Thus, although the assumptions of the simplest motional models that we employ are not expected to strictly hold for spin-labeled lipids in lipid bilayers, the reported agreement with the analyzed

DNP data does not justify the construction of more complex motional models, which would bring additional fitting parameters.

With these considerations in mind, we evaluated the suggestion of the reviewer to separate the reported work into two publications. We believe that two separate papers would be justified only if the theoretical developments and the experiments were comprehensive and exhaustive on their own. For the theory this could mean that, in addition to the free, isotropic tumbling that we have considered, we also develop equations for restricted, anisotropic diffusion, as treated in the MOMD and SRLS models of the Freed lab, for example. Similarly, a stand-alone analysis of experiments could also report cw-EPR and DNP data at other magnetic fields, e.g., X band or W band, as suggested by Gunnar Jeschke (see below). By simultaneously fitting to such experiments at several different magnetic fields, one could then afford to introduce more complex motional models with more fitting parameters. While such comprehensive studies will certainly provide valuable information about the spin-label dynamics in lipid bilayers, they clearly fall beyond the purpose of the current paper. From that perspective, we believe that the reported work does not justify two separate publications.

I suggest to accept this paper as it is with the following minor remarks:

- Already in the abstract states that "the dipolar interaction between the polarized nuclear spins and the polarizing electrons is not completely averaged out by molecular diffusion." But in the case of isotropic rotation, which is considered in the paper, the mean value of the dipole interaction is zero, whether the rotation is slow or fast. It is necessary to explain in what sense the dipole interaction is not completely averaged out. Also I did not find (perhaps I did not notice) the criterion of "slowness" of rotation or tumbling. Perhaps dipole interaction is not averaged to zero in nitroxide-labeled lipids in fluid lipid bilayers, where there is preferred direction in the bilayer, but the paper considers, as I understood, isotropic rotation. Further, if the rotation is slow in the sense of adiabatic mode (the criterion of adiabaticity is known), it may be easier to consider SE in adiabatic approximation with solid angle $\Omega$ as an adiabatic parameter.

  Indeed, we have not specified the condition for solid-effect DNP in liquids. The criterion that we arrived at in our first paper on the solid effect in liquids [2], which had been stated previously by Korringa et al. [3], is that the dipolar interaction should not be completely averaged out *on the time scale of the electronic* $T_2$. Very conveniently, the same time scale dictates whether the tumbling is "slow" or "fast" for the purposes of cw-EPR.

  To address the deficiency identified by the reviewer, we have now modified the first two sentences of the abstract as follows:

  > In spite of its name, the solid effect of dynamic nuclear polarization (DNP) is operative also in viscous liquids, where the dipolar interaction between the polarized nuclear spins and the polarizing electrons is not completely averaged out by molecular diffusion on the time scale of the electronic spin-spin relaxation time. Under such slow-motional conditions, it is likely that the tumbling of the polarizing agent is similarly too slow to efficiently average the anisotropies of its magnetic tensors on the electronic $T_2$ time scale.

  We have also included similar references to the time scale $T_2$ of the electronic spins on several other occasions throughout the manuscript.

- The second minor remark. I was confused by the statement: "In (59), the sum over $l$ mixes only expansion coefficients whose values $l$ differ by two from $L$". Does this mean that $l = L+2$ or $l = L - 2$, rather than the correct condition $|L - 2| \leq l \leq |L + 2|$ ?

  This statement was not only confusing but also incorrect. We modified the sentence as follows:

  > In (59), the sum over $\ell$ mixes only expansion coefficients with $\ell = L, L \pm 2$ [4] because . . .

  In this paragraph we basically repeat the observations that have been made by Freed in [4]. (It is indeed true that $\ell = L \pm 1$ do not occur because the corresponding Clebsch-Gordan coefficients vanish.)

**3 Reviewer 1: Gunnar Jeschke**

Characteristically, Gunnar Jeschke has not only carefully identified the weak and poorly justified parts in our presentation, but has also generously pointed out how these can be addressed and developed. We thus feel fortunate that he has taken the time to review our paper.

This work extends previous theory on solid-effect DNP in viscous liquids by one of the authors to the more complicated situation where g anisotropy is partially averaged by slow tumbling. The derivations provide the basis for a computationally affordable numerical treatment. Field profiles of DNP enhancements ("DNP spectra") fitted on the basis of this approach are in rather good agreement with experiments. The fitted parameters take physically reasonable values, despite the fact that the model for spin label dynamics is strongly simplified with respect to expected dynamics in the experimentally studied systems. The results constitute a substantial advance in a field of current interest. The manuscript is mostly clear and well written. I recommend publication after minor revision that takes into account the following issues.

1. The treatment of the experimental spectra assumes isotropic rotational diffusion (stated, a bit late, in line 602). For 10-Doxyl-PC in DOPC bilayers, this motional model is a strong simplification. Even for 16-Doxyl-PC, which is closer to the chain end and experiences less of an orienting potential, I am not sure how good an isotropic-motion model is. This, and the worsened agreement upon including nitrogen hyperfine coupling/anisotropy for 10-Doxyl-PC (Figure 4A) may indicate error compensation between deficiencies of the motion model and the neglect of $^{14}$N hyperfine coupling. In my opinion, this possibility should be mentioned. The difference in parameters between 10-Doxyl-PC and 16-Doxyl-PC (Table 2) might partially be due to the motional model being worse for 10-Doxyl-PC than for 16-Doxyl-PC.

   To warn the reader that the manuscript treats only free, isotropic rotational diffusion, we have now included the following sentences to the Introduction:

   > For the illustrative purposes of the current paper, we consider only free (i.e., unrestricted) rotational diffusion with an isotropic diffusion coefficient. Nevertheless, the treatment can be analogously extended to anisotropic diffusion in an orienting potential by building on the general mathematical formalism of the MOMD and SRLS models [5, 6].

   A similar clarifying statement was added at the very beginning of Sec. 4. Additionally, we changed the name of Sec. 4.1 from "Stochastic Liouville equation" to "Stochastic Liouville equation for isotropic tumbling".

The comment of the reviewer raises two separate issues: (i) the *isotropic diffusion* model of tumbling and (ii) the *absence of hyperfine interaction* with $^{14}$N on the nitroxide spin label. Here we will explore the second issue, and will address the first issue when engaging with the next remark of the reviewer.

In the manuscript, the fits to the experimental cw-EPR spectra were performed without any hyperfine interaction. To assess the effect of the hyperfine interaction, we then simulated EPR spectra using the parameters from these fits and an axial hyperfine tensor with fixed components: $(14, 14, 90)$ MHz. In the case of 10-Doxyl-PC, the agreement with experiment became (somewhat) worse when the hyperfine interaction was included (fig. A4a). In the case of 16-Doxyl-PC, the hyperfine interaction broadened the negative part of the derivative spectrum (i.e., the side at positive offsets), which was more similar to the experiment than the simulation without the hyperfine interaction. Although all simulated EPR *derivative* spectra (with or without the hyperfine interaction) deviated from the experiment, we observed that the differences of the *integrated* spectra (figs. A4b and A5b) were smaller than the frequency resolution of our DNP field profiles (figs. 1c and 1d). Since the contributions of both the Overhauser and solid effects to the DNP spectrum are expected to be *broadened* versions of the *integrated* EPR spectrum, we reasoned that the quality of our fits to cw-EPR should be sufficient for the purposes of fitting the DNP field profiles. Nevertheless, to explore the possibility of error compensation that is suspected by Gunnar Jeschke, we performed the following additional fits to the cw-EPR spectra.

**10-Doxyl-PC**

(a) Because the fit of $g$ was performed without $A$ (fig. A4a), we thought that including the hyperfine interaction after such fit can only make the agreement with experiment worse. We therefore decided to perform the original fit in the *presence* of $A$ using Easyspin [7]. We thus fitted the components of $g$ and the rotational time scale $\tau$ in the presence of fixed $A = (14, 14, 90)$ MHz. The result is shown on the next page, in the top left corner of fig. 1 (blue line). (The values of the fitted parameters are given in Table 1.) We then removed $A$ and performed a simulation (not a fit!) using the values of the parameters from the fit (red line). To our surprise, the simulation without $A$ was still (somewhat?) better than the fit with fixed $A$.

(b) After that, we decided to also fit the hyperfine tensor $A$. Starting with an isotropic $A$, we performed a fit to the experiment allowing all values of $g$, the single value of $A$, and $\tau$ to vary. The result is shown in the upper right corner of fig. 1 (blue line). (The fitted parameters are again given in Table 1.) In this case, removing the hyperfine interaction (blue line) had almost no effect on the simulated spectrum.

(c) We then fitted an axial $A$ (lower left corner of fig. 1) and a completely anisotropic $A$ (lower right corner of fig. 1), in addition to $g$ and $\tau$. In both cases, the fitted $A_z$ was smaller than $A_y$ (last two rows of Table 1). However, all three spectra with fitted $A$ are seen to be practically identical to each other, which indicates that the experimental spectrum does not contain sufficient information to reliably resolve the three components of $A$.

Encouragingly, all performed fits resulted in essentially identical values of $g$ (Table 1), which are also the values that were used in the manuscript (eq. (97)). Similarly, all simulations in which $A$ was also fitted (last three rows of Table 1) resulted in $\tau \approx 5.2$ ns, which is also the value used in the manuscript.

[Figure]

Figure 1: The experimental cw-EPR spectrum of 10-Doxyl-PC (orange) is *fitted* with a hyperfine tensor (blue) and *simulated* without a hyperfine tensor (red) using Easyspin [7]. The simulation parameters are given in Table 1.

Table 1: Parameters obtained from the fits to the experimental spectrum of 10-Doxyl-PC.

|                     | $A_x, A_y, A_z$ (MHz) | $g_x, g_y, g_z$          | $\tau$ (ns) |
|---------------------|:---------------------:|:-----------------------:|:-----------:|
| fixed $A$           | $14, 14, 90$          | $2.0075, 2.0055, 2.0022$ | 5.7         |
| fitted isotropic $A$| $22, 22, 22$          | $2.0075, 2.0055, 2.0023$ | 5.3         |
| fitted axial $A$    | $24, 24, 11$          | $2.0075, 2.0055, 2.0023$ | 5.2         |
| fitted rhombic $A$  | $0, 31, 11$           | $2.0075, 2.0055, 2.0022$ | 5.2         |

The fact that all three spectra with fitted $A$ practically did not change when $A$ was removed from the simulation demonstrates that the absence of hyperfine interaction is not the origin of the remaining (small) discrepancy with the experiment. Although there could be many different reasons for this discrepancy, it is astonishing how good the fit is considering the overly simple motional model that we have used (i.e., isotropic rotational diffusion).

It could be disturbing at first sight that fitting $A$ with two components (axial) or three components (rhombic) resulted in spectra that are practically the same as the fit with a single component (isotropic). The reason could be that the hyperfine tensor is partially

averaged by fast librations on ps time scales, which are not included in our motional model. Such fast motions could also explain the difference between our fitted $g_x = 2.0075$ and the value of $g_x = 2.00887$ reported for 10-Doxyl-PC in DMPC lipid bilayers containing 40 mol % Cholesterol [8]. (We return to these published experiments when addressing the next point of the reviewer.)

**16-Doxyl-PC**

We performed the same analysis as above also for 16-Doxyl-PC, with the difference that now the three components of the $g$ tensor were fixed at the values determined from the fits of 10-Doxyl-PC. (The same reasoning was followed in the manuscript since the experimental EPR spectrum of 16-Doxyl-PC was not expected to contain sufficient information to determine the components of the $g$ tensor.) The resulting spectra are shown in fig. 2 and the corresponding parameters are given in Table 2.

[Figure]

Figure 2: The experimental cw-EPR spectrum of 16-Doxyl-PC (orange) is *fitted* with a hyperfine tensor (blue) and *simulated* without a hyperfine tensor (red) using Easyspin [7]. The simulation parameters are given in Table 2.

As above, we see that fitting $A$, rather than keeping it fixed at $(14, 14, 90)$ MHz, improves the agreement with experiment (blue lines). Furthermore, the fit with isotropic $A$ does not get any better (or worse) if we fit two (axial) or three (rhombic) values of $A$.

Table 2: Parameters obtained from the fits to the experimental spectrum of 16-Doxyl-PC.

| | $A_x, A_y, A_z$ (MHz) | fixed $g_x, g_y, g_z$ | $\tau$ (ns) |
|---|---|---|---|
| fixed $A$ | $14, 14, 90$ | $2.0075, 2.0055, 2.0022$ | $1.80$ |
| fitted isotropic $A$ | $38, 38, 38$ | $2.0075, 2.0055, 2.0023$ | $1.73$ |
| fitted axial $A$ | $38, 38, 36$ | $2.0075, 2.0055, 2.0023$ | $1.73$ |
| fitted rhombic $A$ | $64, 0, 61$ | $2.0075, 2.0055, 2.0022$ | $1.71$ |

In all cases, we see that repeating the simulation without any $A$ (red lines), while keeping $\tau$ unchanged, leads to a visibly narrower spectral line, which deviates more substantially from the experimental spectrum.

The last three fits with fitted $A$ are seen to match the experiment rather well. They all yield $\tau = 1.7$ ns for the time scale of rotational diffusion. In the manuscript we used $\tau = 1.9$ ns. This slower tumbling must partly compensate for the absence of $A$ by causing some additional broadening. Evidently, there is some error compensation in the case of 16-Doxyl-PC, as suspected by Gunnar Jeschke.

To conclude, we stress again that our aim is to explain the DNP enhancements of 10- and 16-Doxyl-PC. As evident from figs. 5b and 6b in the manuscript, these DNP spectra are available at a much lower resolution along the horizontal (frequency) axis compared to the resolution of the EPR spectra. While the derivative EPR spectrum is known to be extremely sensitive to the details of the motion, its integral is more forgiving (see figs. A4b and A5b). Since the Overhauser and solid-effect components of these DNP spectra are going to be broadened versions of the integrated cw-EPR line shapes, we think that further improving the fit to the cw-EPR spectra is not really justified for the purposes of fitting the DNP spectrum.

2. Motion of spin-labelled lipids in lipid bilayers is a rather well researched topic (see, e.g., Livshits, V.A., Kurad, D., Marsh, D. Multifrequency simulations of the EPR spectra of lipid spin labels in membranes. J. Magn. Reson. 2006, 180, 63-71 and references therein). I believe that the motional model of 16-Doxyl-PC and 10-Doxyl-PC in DOPC bilayers could be substantially improved by measuring CW EPR spectra at a few more frequencies and fitting them by the microscopic-order macroscopic-disorder model (MOMD) developed by Freed et al. While this is clearly beyond the scope of the present study, I would like to see some comment whether the approach introduced in this manuscript could be used with a MOMD model or at least with a model of anisotropic rotational diffusion (axial rotational diffusion tensor).

Again there are two different questions that we need to address: (i) Is there any justification in the literature for the simple motional model of isotropic rotation that we have used for 10- and 16-Doxyl-PC? (ii) Can the formalism that we present be extended to anisotropic diffusion and more complex motional models like MOMD?

**Comparison with literature**

Regarding the first question, the surprisingly good correspondence between the fitted and the experimental EPR spectra in figs. 1 and 2 above, suggests that the extremely simple model of isotropic rotational diffusion that we used is, in fact, acceptable for our purposes. Nevertheless, to address some of the concerns of the reviewer, we checked whether such

simple treatment is supported by the more extensive experimental and computational cw-EPR studies in the literature.

Following the reference given by Gunnar Jeschke [8], we identified two sets of studies of potential relevance. The first set comprises the work of Derek Marsh and colleagues [8, 9, 10] and the second set that of Jack Freed and colleagues [11, 12, 13].

(a) In the first set, the system that has been characterized with multifrequency cw-EPR at 94 and 9 GHz (as well as 34 GHz in [10]) consists of **DMPC** lipid bilayers that contain **40 mol % cholesterol** (as well as 5 mol % cholesterol in [10]). One very attractive feature of this work is that spin labels at all **positions from 4 to 14** on the acyl chain of the labeled lipids were characterized. (Only positions 5, 7 and 9 in the case of 5 mol % cholesterol in [10].) This range covers our position 10 but does not cover our position 16.

(b) The system in the second set of studies (more precisely, the one that is relevant for us) consists of lipid bilayers composed of pure **DPPC** lipids. While [11] is a multifrequency cw-EPR study at 250 and 9 GHz, [12] is a 2D-ELDOR spectroscopy study (at 17 GHz). Unfortunately, all these studies use only spin-labeled lipids labeled at **position 16**.

Depending on the temperature and cholesterol content, lipid bilayers exhibit a variety of different phases: *gel* phase, *liquid-crystalline* phase (LC), and *liquid-ordered* phase (LO). The gel phase, where the lipids are very ordered, occurs at low temperatures. Above the phase transition temperature $T_m$, a pure lipid bilayer (i.e., without cholesterol) is in the LC phase, where the ordering of the individual lipids is reduced. Even at temperatures above $T_m$, the ordering can be increased by introducing cholesterol. At cholesterol concentrations beyond 30 mol % the lipids are in the LO phase [11].

In Table 3 we have summarized the relevant findings from [8, 10] and [11, 12]. Most importantly, we see that Marsh and colleagues have characterized either the gel or the LO phase. However, the DOPC lipid bilayers in our experiments are in the LC phase ($T_m = -17°C$ for pure DOPC). Only the experiments from the Freed lab shed light on the LC phase. (It should be kept in mind that the acyl chains of the DOPC lipids consist of 18 carbon atoms and contain one double bond, whereas the chains of DPPC used in the experiments of Freed et al. are saturated and contain 16 carbon atoms. The chains of DMPC are even shorter, with only 14 carbons.)

Table 3: Summary of the literature. The gel-to-LC phase transition temperature is $T_m = 24°C$ (pure DMPC) and $T_m = 41°C$ (pure DPPC). The order parameter $S_0$, which is restricted to the interval $[0, 1]$, reflects the ordering with respect to the normal of the lipid bilayer. ($S = 0$ implies no ordering.) The time scales $\tau_\perp$ and $\tau_\parallel$ are calculated from the reciprocals of the components of the rotational diffusion tensor, which is assumed to be axially symmetric.

| reference | label | lipid | mol % chol | $T$ (°C) | phase | $S_0$ | $\tau_\perp$ (ns) | $\tau_\parallel$ (ns) |
|---|---|---|---|---|---|---|---|---|
| [8] (Table 4) | 10 | DMPC | 40 | 30 | LO | 0.67 | 112 | 16 |
| [10] (Table 2) | 9 | DMPC | 5 | 10 | gel | $\sim 0.7$ | | |
| [11] (Fig. 3) | 16 | DPPC | 0 | 45 | LC | $< 0.2$ | 0.8 | 2.5 |
| [12] (Table 1) | 16 | DPPC | 0 | 50 | LC | 0.16 | 4 | 0.13 |

The 2D-ELDOR measurements in [12] (as well as the cw-EPR at 250 GHz in [11]) were modeled with the MOMD model [5], which allows for an orienting potential that tends to

align the lipid chains. The degree of alignment is quantified by the order parameter $S_0$, which we have included in Table 3. ($S_0 \lesssim 1$ corresponds to high ordering and $0 \lesssim S_0$ corresponds to low ordering. Free rotational diffusion, i.e., without any ordering, corresponds to $S_0 = 0$.) From the table we see that the ordering is high for the gel and LO phases, but is relatively low for the LC phase. Nevertheless, it is non-zero even for 16-Doxyl-PC in the LC phase of DPPC. Although 10-Doxyl-PC is expected to be even more ordered than that, it is not clear exactly to what extent.

In conclusion, we see that the model of unrestricted rotational diffusion that we have employed is not strictly correct even for 16-Doxyl-PC. However, it is not as bad as one would expect in the light of studies characterizing the gel or LO phases of lipid bilayers.

The MOMD modeling in [11, 12] has been performed for rotational diffusion with an axial tensor. The reciprocals of the two fitted values of this diffusion tensor are given in the last two columns of Table 3. Although the larger and smaller of the two components of the axial diffusion tensor have come out differently in [11] and [12], it is encouraging to see that, overall, the time scale of the MOMD modeling is very similar to the time scale that we deduced from free, isotropic diffusion ($\tau = 1.9$ ns for 16-Doxyl-PC). (Since [11, 12] use $g_x = 2.0089$, it is likely that the faster, sub-ns time scale of the axial diffusion achieves the partial averaging of the $g$ tensor, thus bringing the effective $g_x$ closer to our fitted value of $g_x = 2.00755$.)

In the light of this comparison with the literature, we conclude that a MOMD model with an axial diffusion tensor will likely improve our fits to the experimental cw-EPR spectra of both 16-Doxyl-PC and 10-Doxyl-PC. However, as already mentioned before, the small discrepancies in the cw-EPR spectra that our very simple modeling fails to account for are likely to be insignificant once the effects of power-broadening and motional broadening are included when simulating the DNP field profiles.

**Anisotropic diffusion and MOMD**

While we believe that the experimental uncertainty of our DNP field profiles (figs. 1c and 1d in the manuscript) does not justify more complex modeling with anisotropic and restricted rotation, the question of whether the presented theoretical formalism is, in principle, extendable to models like MOMD is highly relevant. Since we feel that this points deserves to be mentioned in the manuscript, **we have now included a new section 5.4 titled** "Limitations of the modeling", where we also discuss the issue identified by Gunnar Jeschke in point 3 below.

3. I wonder how difficult it would really be to extend the approach to including $^{14}$N hyperfine anisotropy. I would assume that computation time increases only by a factor of three, because the subspaces with different nuclear magnetic quantum number of $^{14}$N could be treated independently. The g and hyperfine tensor being coaxial, each subspace could be computed with an effective tensor. The authors may want to consider whether this would be feasible and comment on it. Again, it may be beyond the scope of this study, but may also be of interest, as neglect of $^{14}$N hyperfine coupling does make a difference (Figures A4, A5).

The difficulty of including the hyperfine interaction is not necessarily computational but rather analytical. At present, we identify the polarizing agent with an $S = 1/2$ electronic spin, which allows us to describe the spin dynamics (in Liouville space) with the classical Bloch equations. If we were to include hyperfine coupling to $^{14}$N ($I = 1$), the dimension of the Liouville space would need to increase by a factor of nine. Computationally, this increase

of dimension is not a problem, as we know from the efficient slow-motional simulations of the Freed lab. In fact, the 1971 paper that we have followed in our presentation [4], shows how to account for (the secular and pseudosecular parts of) the hyperfine interaction of a nitroxide.

The difficulty in our case is that we previously formulated the spin dynamics of relevance to the solid effect in terms of two successive sets of Bloch equations [1]. Therefore, to be able to include hyperfine coupling to $I = 1$ the way it is done in slow-motional EPR, we would also need to similarly reformulate the second set of "Bloch equations" in the solid effect. Ultimately, if the theory is to be applied to solid-effect DNP with nitroxide polarizing agents at X, Q or W band, both sets of "Bloch equations" should be re-expressed in Liouville space. However, given that the solid-effect DNP enhancements with nitroxides are rather small, including at X band [14], it is not clear whether such effort is presently justified.

From this perspective, it must be clear that the J-band experiments that we analyze in the manuscript were carefully selected exactly because the hyperfine interaction was expected to be negligible compared to the anisotropy of the g tensor at this high magnetic field, thus allowing us to illustrate the theoretical formalism in its simplest possible form.

4. Work such as this one is notoriously hard to reproduce without having access to the programs used for the computations. The authors may want to consider putting the program on GitHub.

In line with the publishing requirements of *Magnetic Resonance*, sharing the code has been our intention from the beginning. Its latest version can be accessed at
`https://github.com/dzsezer/solidDNPliquids_g-tensor`.

Details:

1. Around line 110: "cw-EPR spectrum deviates from a Lorentzian line". I would have written "cw-EPR lineshape is not Lorentzian", because spectra often consist of several Lorentzian lines, not only one.

The suggested modification was implemented.

2. When first discussing the deviation in Fig. 1(d) between experimental data and the decomposition into OE and SE contribution, it would be helpful to tell the reader that the paper addresses this issue.

We have now included the following sentence after the first discussion of the deviations:

> The theory presented below (Sec. 4) aims to address these deficiencies of the simple approach.

3. Line 201, "When generalizing the Bloch equations to non-isotropic g tensor" I would have written "When generalizing the Bloch equations to an anisotropic g tensor".

The suggested change was implemented.

4. Line 220/221: "For the same analysis to apply to solids, spin diffusion should be much faster". I do not understand how spin diffusion enters the picture here. In the previous sentence, you mention the correlation time of the (supposedly electron-nuclear) dipolar interaction. There is no electron-nuclear spin diffusion that could be an analogous phenomenon in solids.

In solids, for a given nucleus, the electron-nucleus dipolar interaction does not change with time since the spatial positions of the two spins remain fixed. (Here we think about static

sample, i.e., without MAS.) In this picture where one focuses on the fixed positions of the nuclei, nuclear spin diffusion can be visualized as the simultaneous flip in opposite directions of two nuclear spins at nearby sites. Clearly, such flip-flop transition changes an up (down) spin at site 1 to a down (up) spin and a down (up) spin at site 2 to an up (down) spin.

Rather than "sitting" on one site and flipping the spin on this site, the same process of nuclear spin diffusion can be viewed as the jump of an up (down) spin from site 1 to site 2, with the concurrent jump of a down (up) spin from site 2 to site 1. In this alternative view, the nuclear spin that happens to be at site 1 at some give time will gradually "diffuse" to other sites at later times. It is this second picture of nuclear spin diffusion that we have in mind when applying the mathematical formalism of liquids to the solid state. Clearly, by following the diffusion of an up (down) nuclear spin over many different nuclear sites, we also end up with an electron-nucleus spin diffusion of the type that we have modeled with the FFHS model.

In an effort to clarify this point we have now modified the relevant sentences in the manuscript as follows:

> In liquids, $A_1$ changes in time because of molecular diffusion. The treatment of SE-DNP for time-dependent $A_1$ in [2] was developed under the assumption that the nuclear $T_1$ is orders of magnitude larger than the correlation time of the electron-nucleus dipolar interaction, which is practically always the case in liquids. For the same analysis to apply to solids, nuclear spin diffusion, which analogously to molecular diffusion in liquids spreads out the nuclear polarization across the sample, should be much faster than the nuclear $T_1$. Although this condition is not necessarily satisfied in the solid state, for the mathematical description in terms of a dipolar correlation function to apply, we will assume that spin diffusion is fast when referring to solids.

5. Line 224/225: "correlation time of the dipolar interaction is infinitely long (but still much shorter than the nuclear $T_1$" Please replace "infinitely long" with "much longer than ...". Probably you compare to relaxation on zero-quantum and double-quantum transitions in the absence of fluctuating electron-nuclear dipolar interaction?

We have now modified the statement as follows:

> Similarly, when accounting for g-tensor anisotropy below, we will assume that the tumbling of the radical is much faster than the nuclear $T_1$. This assumption is clearly violated in solids where "tumbling" is infinitely slow. Nevertheless, for the purposes of comparison, we will refer in the following to 'solids' with the understanding that the correlation time of the dipolar interaction is much shorter than the nuclear $T_1$ (in order to treat nuclear spin diffusion on the level of a translational correlation function) but much longer than all other relaxation time scales (in order to treat the electron-nucleus dipolar interaction as constant). Because we will keep all other parameters, including the time scale of radical tumbling, the same when comparing 'solids' and liquids, it should be kept in mind that our treatment is not a good model for the solid state (hence the quotation marks).

6. Figure A4: The simulation agrees better with experiment without considering the hyperfine interaction that certainly does exist. Some error compensation must be at play here. This needs to be mentioned.

We now have the following sentences in the new section 5.4, which discusses the limitations of the modeling:

Figures A4 and A5 show our attempt to assess the contribution of the neglected hyperfine interaction to the (integrated) EPR spectra of 10- and 16-Doxyl-PC. A somewhat more detailed analysis is contained in our response to the reviewers, which is freely accessible online. There we observe that the hyperfine interaction slightly broadens the EPR line of 16-Doxyl-PC (which is also visible in fig. A5a). Since the only mechanism of broadening in our case is the rotational tumbling, our choice of $\tau_{\mathrm{rot}} = 1.9\,\mathrm{ns}$ likely compensates for some of the "missing" hyperfine broadening. Such compensation does not appear to be happening in the case of 10-Doxyl-PC, where the hyperfine interaction changes the shape but not the width of the EPR line (fig. A4a).

To summarize, in addition to the clarifying sentences inserted on several places throughout the manuscript, the major modification has been the **entirely new section 5.4** which discusses the three central limitations of our treatment. (These are the absence of hyperfine interaction, the absence of orienting potential, and the simplified treatment of translational diffusion.) In this new section, we briefly outline how one could go about addressing each deficiency, and also speculate on why the simplified analysis could still be acceptable in our case.

The first paragraph of the Conclusion is also new.

**References**

[1] D. Sezer, "Non-perturbative treatment of the solid effect of dynamic nuclear polarization," *Magn. Reson.*, vol. 4, no. 1, pp. 129–152, 06 2023. [Online]. Available: https://mr.copernicus.org/articles/4/129/2023/

[2] ——, "The solid effect of dynamic nuclear polarization in liquids," *Magn. Reson.*, vol. 4, no. 1, pp. 153–174, 06 2023. [Online]. Available: https://mr.copernicus.org/articles/4/153/2023/

[3] J. Leblond, J. Uebersfeld, and J. Korringa, "Study of the liquid-state dynamics by means of magnetic resonance and dynamic polarization," *Physical Review A*, vol. 4, no. 4, pp. 1532–1539, 10 1971. [Online]. Available: https://link.aps.org/doi/10.1103/PhysRevA.4.1532

[4] J. H. Freed, G. V. Bruno, and C. F. Polnaszek, "Electron spin resonance line shapes and saturation in the slow motional region," *The Journal of Physical Chemistry*, vol. 75, no. 22, pp. 3385–3399, 10 1971. [Online]. Available: https://doi.org/10.1021/j100691a001

[5] E. Meirovitch, A. Nayeem, and J. H. Freed, "Analysis of protein-lipid interactions based on model simulations of electron spin resonance spectra," *The Journal of Physical Chemistry*, vol. 88, no. 16, pp. 3454–3465, 08 1984. [Online]. Available: https://doi.org/10.1021/j150660a018

[6] A. Polimeno and J. H. Freed, "Slow motional esr in complex fluids: The slowly relaxing local structure model of solvent cage effects," *The Journal of Physical Chemistry*, vol. 99, no. 27, pp. 10 995–11 006, 07 1995. [Online]. Available: https://doi.org/10.1021/j100027a047

[7] S. Stoll and A. Schweiger, "Easyspin, a comprehensive software package for spectral simulation and analysis in epr," *Journal of Magnetic Resonance*, vol. 178, no. 1, pp. 42–55, 2006. [Online]. Available: https://www.sciencedirect.com/science/article/pii/S1090780705002892

[8] V. A. Livshits, D. Kurad, and D. Marsh, "Simulation studies on high-field epr spectra of lipid spin labels in cholesterol-containing membranes," *The Journal of Physical Chemistry B*, vol. 108, no. 27, pp. 9403–9411, 07 2004. [Online]. Available: https://doi.org/10.1021/jp035915p

[9] D. Kurad, G. Jeschke, and D. Marsh, "Lateral ordering of lipid chains in cholesterol-containing membranes: High-field spin-label epr," *Biophysical Journal*, vol. 86, no. 1, pp. 264–271, 2023/09/08 2004. [Online]. Available: https://doi.org/10.1016/S0006-3495(04)74102-8

[10] V. A. Livshits, D. Kurad, and D. Marsh, "Multifrequency simulations of the epr spectra of lipid spin labels in membranes," *Journal of Magnetic Resonance*, vol. 180, no. 1, pp. 63–71, 2006. [Online]. Available: https://www.sciencedirect.com/science/article/pii/S1090780706000127

[11] Y. Lou, M. Ge, and J. H. Freed, "A multifrequency esr study of the complex dynamics of membranes," *The Journal of Physical Chemistry B*, vol. 105, no. 45, pp. 11 053–11 056, 11 2001. [Online]. Available: https://doi.org/10.1021/jp013226c

[12] A. J. Costa-Filho, Y. Shimoyama, and J. H. Freed, "A 2d-eldor study of the liquid ordered phase in multilamellar vesicle membranes," *Biophysical Journal*, vol. 84, no. 4, pp. 2619–2633, 2003. [Online]. Available: https://www.sciencedirect.com/science/article/pii/S000634950375067X

[13] Y.-W. Chiang, Y. Shimoyama, G. W. Feigenson, and J. H. Freed, "Dynamic molecular structure of dppc-dlpc-cholesterol ternary lipid system by spin-label electron spin resonance," *Biophysical Journal*, vol. 87, no. 4, pp. 2483–2496, 2004. [Online]. Available: https://www.sciencedirect.com/science/article/pii/S0006349504737200

[14] B. Gizatullin, C. Mattea, and S. Stapf, "Molecular dynamics in ionic liquid/radical systems," *The Journal of Physical Chemistry B*, vol. 125, no. 18, pp. 4850–4862, 05 2021. [Online]. Available: https://doi.org/10.1021/acs.jpcb.1c02118